# Human brain representations of internally generated outcomes of approximate calculation revealed by ultra-high-field brain imaging

Sébastien Czajko [1,2], Alexandre Vignaud[3] & Evelyn Eger [1] ✉

Much of human culture's advanced technology owes its existence to the ability to mentally manipulate quantities. Neuroscience has described the brain regions overall recruited by numerical tasks and the neuronal codes representing individual quantities during perceptual tasks. Nevertheless, it remains unknown how quantity representations are combined or transformed during mental computations and how specific quantities are coded in the brain when generated as the result of internal computations rather than evoked by a stimulus. Here, we imaged the brains of adult human subjects at 7 Tesla during an approximate calculation task designed to disentangle in- and outputs of the computation from the operation itself. While physically presented sample numerosities were distinguished in activity patterns along the dorsal visual pathway and within frontal and occipito-temporal regions, a representation of the internally generated result was most prominently detected in higher order regions such as angular gyrus and lateral prefrontal cortex. Behavioral precision in the task was related to cross-decoding performance between sample and result representations in medial IPS regions. This suggests the transformation of sample into result may be carried out within dorsal stream sensory-motor integration regions, and resulting outputs maintained for task purposes in higher-level regions in a format possibly detached from sensory-evoked inputs.

Understanding and mentally manipulating numbers is a vital skill in our everyday life and is an important foundational capacity for the technological advances of human society. Being impaired in this capacity, as is, for example, the case in developmental dyscalculia, a learning disability characterized by difficulties in acquiring number concepts and learning mental arithmetic in spite of otherwise normal intellectual functioning, can have profound implications for the concerned individuals' quality of life and socio-economic status[1]. Tackling the nature of such deficits and the neuronal basis of the successful development of mathematical abilities requires a sufficiently precise dissection of the neurophysiological mechanisms underlying different facets of numerical processing, which continues to be a challenge within the field of cognitive neuroscience.

Much progress has already been made regarding the relatively coarse system-level scale description of brain function in relation to numerical processing: Several neuroscientific methods ranging from

[1]Cognitive Neuroimaging Unit, INSERM, CEA, CNRS, Université Paris-Saclay, NeuroSpin center, Gif-sur-Yvette, France. [2]EDUWELL team, Lyon Neuroscience Research Centre, INSERM U1028, CNRS UMR5292, Lyon 1 University, Lyon, France. [3]UNIRS, CEA, Université Paris-Saclay, NeuroSpin center, Gif-sur-Yvette, France. ✉e-mail: evelyn.eger@gmail.com

macaque neurophysiology over human functional imaging to lesion studies converge on supporting a role of areas in and around the intraparietal sulcus, as well as of lateral prefrontal regions, for numerical tasks ranging from merely perceiving and detecting changes in numbers via quantitative decisions to more or less complex mathematical operations[2–5]. Rather than merely distinguishing activation for numerical as opposed to non-numerical tasks overall, functional imaging work in humans has proceeded to characterize the representations and neural codes underlying numerical information in human cortex, demonstrating brain signals distinguishing between individual numerical quantities: for different numbers of items in a visual display (typically, a set of dots), and to some extent the quantity represented by number symbols, brain signals show numerical distance dependent adaptation, as well as distinguishable patterns of direct evoked multi-voxel activity in intraparietal as well as some lateral prefrontal brain regions[6–12]. In some of these regions, individual fMRI recording sites' responses are furthermore found to form orderly topographic layouts as a function of the number of items and to express tuning profiles similar to those observed by neurons in similarly located and possibly functionally equivalent regions of macaque monkey cortex[13–15]. More recent studies have further shown that information closely related to the number of items, which is not reducible to responses to other quantitative descriptors of dot sets such as their density, total area, etc., is also available in several earlier-level brain regions upstream from intra-parietal cortex[16–18]. In sum, previous studies suggest that more explicit numerical representations in higher-level (e.g., intra-parietal) brain regions may arise from the gradual transformation of the visual information evoked at earlier sensory processing stages along the dorsal visual pathway.

What the studies reviewed so far have in common is that they are primarily concerned with "perceptual" number representations, that is, neuronal patterns elicited in response to being presented with, and in some cases in addition internally maintaining, a given presented stimulus. To a large extent, these are likely to reflect representations that are part of the sensory extraction of numerical information from visual images. In addition, some recent work has dissociated representations underlying the sensory encoding of numbers of items from those maintaining those same entities in short-term memory, by manipulating which of two presented dot pattern stimuli was to be remembered[19]. The latter kind of representation was detected in intraparietal but not earlier or later cortical regions. However, numbers are not only perceptual objects but also prime examples of entities that we commonly transform internally during mental computations, generating new numbers as outcomes. Such internal number manipulation is not limited to symbols nor dependent on formal mathematical training, as attested by the ability of young children[20–22], subjects from cultures with very limited numerical symbol use[23], and to some extent even macaque monkeys[24], to perform basic numerical operations over sets of objects. In the latter case, the results of the computations are approximate, contrasting with the precise nature of outputs that can be computed during formal arithmetic on symbols. While these studies document the extent to which such abilities exist, we currently know very little about where and how quantities are coded in the brain when generated internally as the result of such mental computations, neither in symbolic nor non-symbolic format. It remains unclear to what extent some of the representations previously described by fMRI studies during perceptual tasks would also underlie the representation of internally generated contents, or not.

The present study aimed at identifying representations of such internally generated numerical contents, by using the enhanced signal-to-noise ratio provided by ultra-high-field functional imaging. We used a specifically designed paradigm based on an approximate calculation task on non-symbolic numerical stimuli (sets of dots), motivated by the observation that non-symbolic quantities evoke brain representations that are more clearly distinguished in fMRI than symbolic ones, e.g.[8,25].

This was combined with an analysis logic which maximized the separability of brain codes internally generated from those of stimulus-evoked contents as well as other partially correlated components of a mental computation task. Similar to fMRI studies on the mental transformation of simpler visual information represented in early visual areas[26], in our task, the result of computation had to be held in working memory over a prolonged delay period to enhance its distinctness from the stimulus-evoked activity present earlier during the trial. We further relied on pattern analyses, allowing us to isolate the unique contributions of several predictors of the stimulus conditions' similarity, as successfully applied previously to disentangle visual perceptual representations of partially correlated quantities[16]. Our results provide a demonstration of purely internally generated representations of numerical contents in the brain, which, in contrast with stimulus-evoked ones, are found predominantly at higher cortical levels on top of or beyond the visual sensory processing hierarchy.

## Results
During 7T fMRI, subjects were engaged in an approximate non-symbolic calculation task (Fig. 1a) requiring them on each trial to visually process a briefly presented sample number (6, 12, 24, or 48 simultaneously presented items) and to perform on this number an operation as specified by a following symbolic cue (multiplication or division by 2 or 4). They then had to keep the resulting quantity in mind up to the appearance of a probe number (another dot set presented 12 seconds after the sample), requiring a smaller-larger decision with respect to the result of the preceding computation. A relatively long delay period was explicitly chosen to account for the slow nature of the hemodynamic response and to allow for the initial stimulus-evoked activity to at least partly return to baseline to facilitate detection of the internally generated result number representation.

### Behavioral results for comparison with probe numbers
Probe numbers presented at the end of the delay period differed from the correct result on each trial by one of 8 possible ratios (see Methods for details). The percentage of correct responses for comparing the probe to the internally computed result (mean ± SEM) was, on average, 76.8 ± 0.02 for division and 73.5 ± 0.01 for multiplication, with the difference not reaching significance ($t(16) = 1.84$, $p = 0.09$, Cohen's $d = 0.42$), suggesting that subjects were able to perform the task reasonably well and equivalently for the two types of operations. Combined across operations, the accuracies for the different comparison ratios were: ratio 0.50: 88.9 ± 0.03, ratio 0.67: 79.2 ± 0.03, ratio 0.80: 68.2 ± 0.04, ratio 0.91: 55.0 ± 0.04, ratio 1.10: 60.4 ± 0.03, ratio 1.25: 75.4 ± 0.03, ratio 1.50: 84.6 ± 0.03, ratio 2: 89.2 ± 0.03. The range of comparison ratios used further allowed us to fit psychometric functions and compute the just noticeable difference (JND) for the comparative judgment performed here. Estimated across all numerosities, the JND (on a logarithmic scale) corresponded to 0.289 ± 0.03 for division and 0.287 ± 0.02 for multiplication. We also tested for the presence of a so-called operational momentum effect (referring to larger or smaller than expected results reported depending on the type of operation performed), which has been described for mental calculation, most prominently in the non-symbolic case[27,28]. To measure the presence of such effects here, we estimated the point of subjective equality (PSE) of the psychometric function separately for trials with division as opposed to multiplication. The expected tendency with division leading to under- and multiplication to overestimated results should lead to the PSE being shifted towards the left or right, respectively, for these operations. Results showed that PSEs were, on average, −0.029 ± 0.06 for division, and −0.062 ± 0.06 for multiplication, with the difference not being significantly different across subjects ($t(16) = 0.32$, $p = 0.75$, Cohen's $d = 0.07$), suggesting that result representations were, on average, unbiased by the type of operation performed in the task used here.

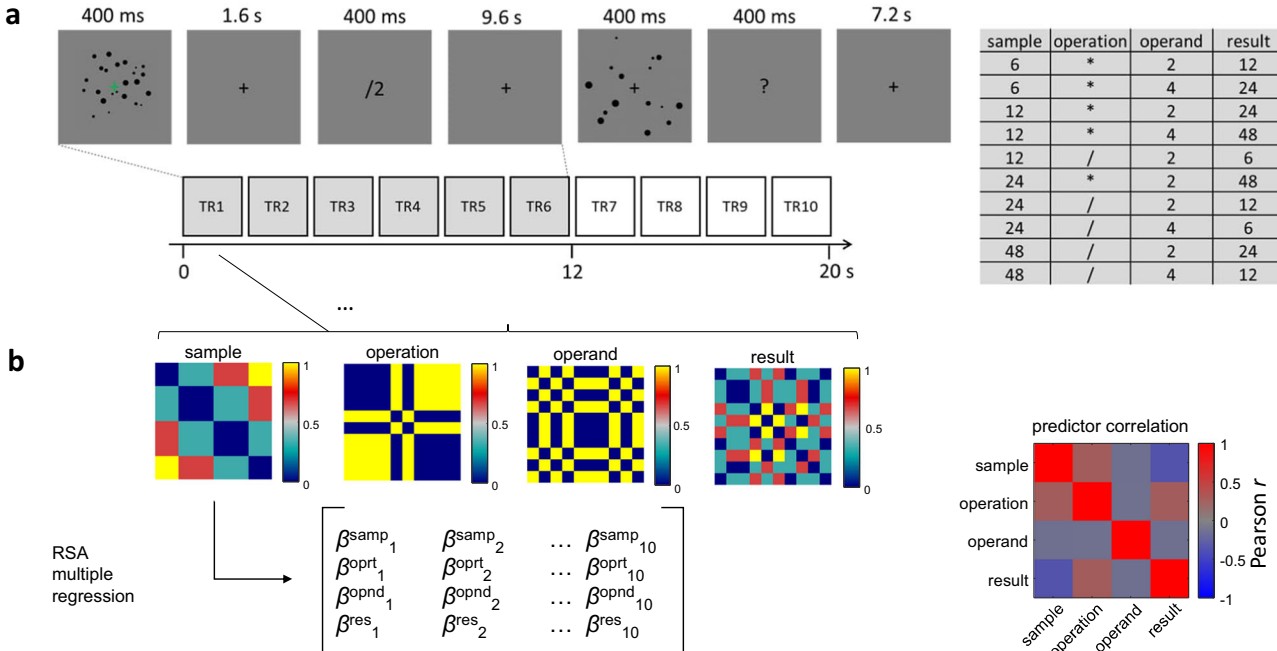

**Fig. 1 | Experimental Paradigm and analytic approach. A** Illustrates an experimental trial where subjects were briefly presented with a cloud of visual items from which they had to extract its approximate numerosity, followed by a symbolic cue instructing them which operation to perform on it. The result of the operation then had to be maintained in memory over a prolonged delay period (9.6 s on standard trials as shown here, randomly shortened on a smaller subset of additional catch trials). At the end of the delay period, another cloud of items appeared on the screen, which required a smaller/larger decision in comparison with the internally maintained result of the operation. The total trial duration (delay between two sample presentations) was 20 s. The 10 experimental conditions used, based on the combination of four different sample numerosities with two operations and two operands, are shown on the right. **B** Illustration of inputs to the pattern analysis employed to disentangle the stimulus-evoked and internally generated representations of quantities from other components of the computation: 4 predictor matrices reflecting the Euclidean distances of these conditions in terms of sample numerosity, operation, operand, and result numerosity (on a logarithmic scale), here color scaled between 0 and 1 for visualization purposes, were entered all together into a multiple regression to predict the fMRI pattern similarity across the 10 conditions for each acquired fMRI data volume (TR corresponds to time of repetition of one volume). The multiple regression yielded estimates of effect sizes ($\beta$) for sample (samp), operation (oprt), operand (opnd), and result (res) predictors. The correlations between individual predictors are shown on the right. While it is generally impossible to entirely decorrelate in- and outputs of calculation tasks from each other and from the operation at the same time, our specific conditions were chosen such that sample and result predictors were roughly equally correlated with the operation predictor, and sample and result predictors were slightly negatively correlated with each other (Pearson $r = -0.34$).

## FMRI pattern analysis to disentangle representations of sample, result, operation, and operand

Our focus in this contribution is on specific fMRI pattern analyses disentangling the unique contribution of each of several predictors (sample numerosity, operation, operand, and result numerosity) to the pattern of observed distances between all possible pairs of experimental conditions in multi-voxel space. All these predictors were applied here in a time-resolved fashion (for activity estimates at multiple time bins corresponding to each acquired image volume); see Fig. 1b. Thus, our chosen analyses make no assumptions on the particular time course of the response for each predictor. Entering all predictors together into a multiple regression at each time point should ensure that the estimated effects only reflect the variance that each predictor can explain on top of all the others at each of these time points, also see[16]. Complementary exploratory analyses using univariate methods relying on a canonical hemodynamic response function are described in the Supplementary Materials, and results are presented in Supplementary Fig. 1 and Supplementary Table 1.

Performing the mentioned fMRI pattern analyses at a local scale within a small searchlight at the participant level, followed by group statistics in surface space, revealed most prominently the (stimulus-evoked) representation of the sample numerosity which involved predominantly dorsal stream intra-parietal, but also some lateral prefrontal and occipito-temporal brain regions, most strongly during middle parts (4–8 s) of the trial delay period (Fig. 2 and Table 1). During the late parts of the delay (8–12 s) before the probe numerosity

appeared on the screen, the different internally generated results could be distinguished in lateral prefrontal regions overlapping, according to Freesurfer's parcellation of major gyri and sulci[29] with the left middle frontal gyrus and the right inferior precentral sulcus, and the left angular gyrus. A last significant cluster was present in the medial parietal lobe (right subparietal sulcus). A slightly more inferior / anterior left lateral prefrontal region (overlapping with the inferior frontal sulcus) was found to distinguish between the two operands, detectable during the same later parts of the delay. No significant effects were detectable for the operation at the level of the searchlight analysis. For completeness, we also performed additional analyses corresponding to differential comparisons between time periods: middle > early, late > early, late > middle, and late and middle > early. Results of these analyses, reported in detail in Supplementary Fig. 2 and Supplementary Table 2, confirmed that sample-related effects were most pronounced during middle compared to early time periods and showed that result-related effects while being most pronounced in the late > early contrast, were already detectable in the left angular gyrus region in the middle > early contrast.

In addition to the searchlight analyses scanning the entire acquired brain volume for local pattern information related to our predictors of interest, we investigated selected regions of interest, focusing specifically on subregions of the parietal and lateral prefrontal cortex. The aims of these analyses were twofold: (1) To investigate information in brain response patterns at a slightly more extended spatial scale than the one corresponding to the radius of the

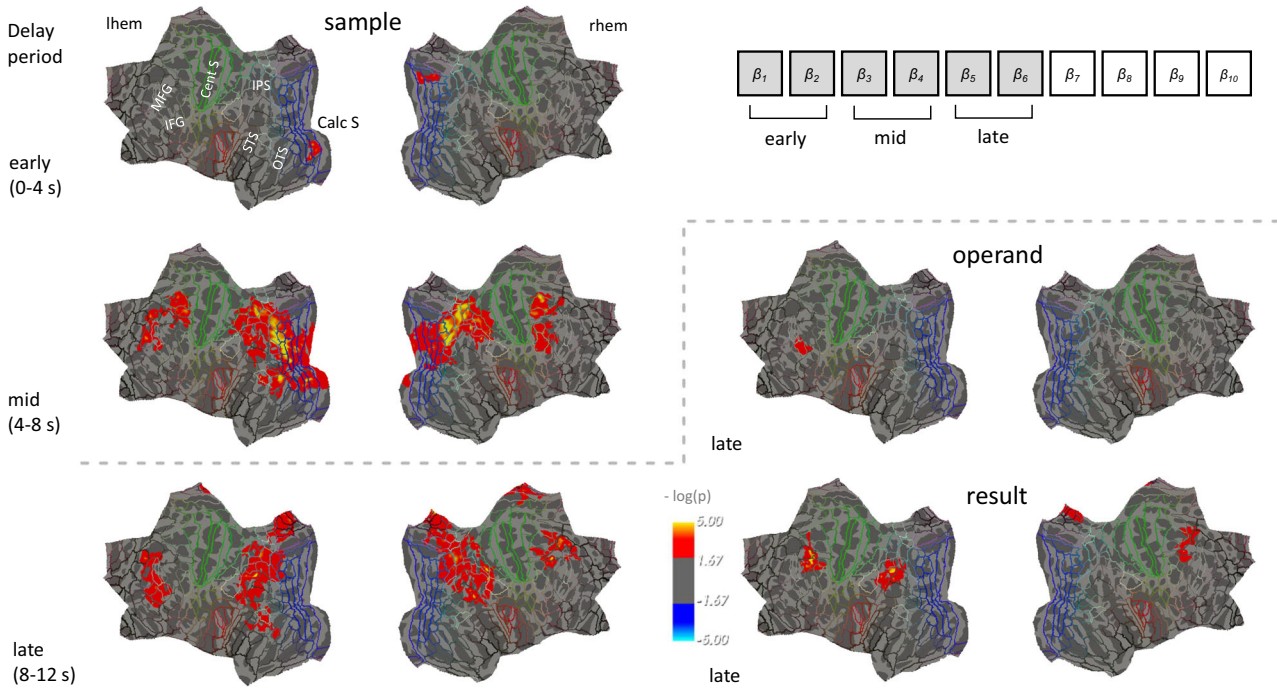

**Fig. 2 | Results of representational similarity searchlight analysis.** Analyses (*n* = 17 subjects) tested for the effects of sample, operation, operand, and result predictors on brain activity pattern dissimilarity in a multiple regression approach. Analyses were initially performed for each subject per time bin/acquired volume during a prolonged delay period where subjects computed the result of an operation and subsequently kept it in memory, with effect size estimates (*β*) then being averaged across early, middle and late parts of the delay period (right) before performing surface-based group analyses in FreeSurfer (two-tailed one-sample t-tests, correction for multiple comparisons by permutation at cluster level,

$p_{FWE} < 0.05$, cluster forming threshold $p < 0.01$). The color scale represents uncorrected −log(*p*) values within the clusters surviving correction. Results are projected onto a flattened cortical average surface (https://mri.sdsu.edu/sereno/csurf/), with superimposed lines representing the regional borders of the Human Connectome Project (HCP-MMP1) parcellation[30]. For better orientation, the location of major anatomical landmarks (MFG middle frontal gyrus, IFG inferior frontal gyrus, Cent S central sulcus, IPS intraparietal sulcus, STS superior temporal sulcus, OTS occipitotemporal sulcus, Calc S calcarine sulcus) is indicated. Source data are provided as a Source Data file.

previous searchlight analysis within functional-neuroanatomically meaningful subdivisions of the cortex, and (2) evaluate the degree of co-localization of sample and result representations at that level, within independently defined regions of interest. A set of 16 subregions of interest, each within the parietal and frontal cortex, were derived from the HCP_MMP1 parcellation[30], which is based on a combination of functional and anatomical criteria using a large-scale multimodal data set (see Fig. 3, and the Methods section for the selected regions). As one additional characterization of these regions, their average BOLD signal time courses for the 10 different experimental conditions in the present paradigm can be seen in Supplementary Figs. 3 and 4, for the parietal and frontal ROIs, respectively. These plots illustrate that while part of them (for example, most of the ROIs around the intra-parietal sulcus) showed strong stimulus-evoked activity, this was less the case in the lateral parietal and most frontal regions.

Results of multivariate pattern analyses, focusing on information in distributed activity patterns rather than overall activation level, showed that information about the sample numerosity was widely distributed in the parietal cortex (Fig. 3a), reaching significance in all subregions in both the middle and late delay periods. Information on the result numerosity was detectable during the late delay period in regions around the angular gyrus (regions PFm, PGi, PGs), confirming the results of the previous searchlight analysis. The ROI analysis, however, also revealed additional significant information on the result in regions within the depth of / along the superior wall of the IPS (region MIP during the middle and late delay period and region IP1 during the late delay period). Within the lateral prefrontal cortex, information on the sample numerosity was widespread and significantly detectable within all subregions tested during both time intervals. The result representation was detectable only during the late

delay period and in most subregions with a few exceptions (Fig. 3b). Overall, these ROI analysis results confirm the tendency of the representation of the sample to be stronger or more widespread than the one of the result, as had already been suggested by the searchlight analysis, and they show that subregions which have significant information about the result do in general represent information about the sample numerosity as well. While our interest here is mostly in the representation of stimulus-evoked and internally generated representations of the individual quantities, we also performed equivalent exploratory ROI analyses on the operation and operand predictors, the results of which can be found in Supplementary Fig. 5.

## FMRI pattern analysis to test for a shared representational space between sample and result numerosities

Multiple regression analysis on representational distance matrices as those described so far can reveal the unique contribution of the result representation disentangled from the other (partly correlated) aspects of the computation, and it can show in how far information on sample and result numerosities is both present in a given part of the cortex. What it cannot reveal, though, is whether and at which level in the brain's cortical hierarchy the perceptually evoked sample representations and the internally generated result representation are encoded within a shared representational space (that is, the neuronal populations allowing to distinguish a specific numerosity as, e.g., 16, from 8 or 32, when perceived as the sample, would also allow the same discrimination when these are computed as the result). We might hypothesize the existence of such a shared representational space, at least within the regions where the transformation from the sample into the result initially occurs. To test for regions behaving in line with this idea, we used a multivariate decoding approach to predict numerosity, not

**Table 1 | Cluster summary table for the surface-based group analysis (_N_ = 17) conducted on the searchlight pattern analysis (multiple regression with sample, operation, operand, and result predictors on the fMRI distance matrices)**

| Hemisphere | Cluster label | Max | Size (mm²) | MNI X | MNI Y | MNI Z | CWP | NVtxs |
|---|---|---|---|---|---|---|---|---|
| _Sample—early delay_ | | | | | | | | |
| Left | G_oc-temp_med-Lingual | 4.53 | 614.0 | −12.5 | −84 | −13.2 | 0.0314 | 609 |
| Right | S_parieto_occipital | 4.45 | 527.6 | 12.8 | −71.2 | 19.4 | 0.0466 | 662 |
| _Sample—middle delay_ | | | | | | | | |
| Left | S_oc_sup_and_transversal | 6.33 | 13320.9 | −34.6 | −80.1 | 19.2 | 0.0008 | 22,980 |
| | S_front_sup | 4.60 | 3222.6 | −20.6 | 29.3 | 33.6 | 0.0120 | 5834 |
| Right | S_oc_sup_and_transversal | 6.03 | 9186.8 | 34.6 | −73.1 | 23.2 | 0.0012 | 17,077 |
| | S_front_sup | 5.22 | 1633.0 | 21.5 | −3.8 | 55.7 | 0.0262 | 3198 |
| | S_precentral-inf-part | 4.46 | 1605.6 | 38.7 | 2.8 | 28.2 | 0.0266 | 3090 |
| _Sample—late delay_ | | | | | | | | |
| Left | G_pariet_inf-Angular | 6.81 | 5010.4 | −41.1 | −64.7 | 45.4 | 0.0032 | 10,617 |
| | S_front_sup | 4.51 | 3045.1 | −19.5 | 40.2 | 33.6 | 0.0068 | 4758 |
| | S_subparietal | 4.20 | 1432.9 | −6.8 | −48.9 | 29.6 | 0.0203 | 2991 |
| | G_temporal_middle | 3.50 | 1006.9 | −55.2 | −56.9 | −2.8 | 0.0314 | 1710 |
| Right | G_pariet_inf-Supramar | 5.37 | 8843.4 | 53.8 | −38.4 | 45.4 | 0.0004 | 19,265 |
| | S_front_sup | 4.61 | 1368.1 | 20.3 | 22.5 | 44.7 | 0.0207 | 2528 |
| | G_front_middle | 5.39 | 880.0 | 41.9 | 16.7 | 43.5 | 0.0427 | 1461 |
| _Operand—late delay_ | | | | | | | | |
| Left | S_front_inf | 3.03 | 608.9 | −37.1 | 30.2 | 12.9 | 0.0270 | 941 |
| _Result—late delay_ | | | | | | | | |
| Left | G_pariet_inf-Angular | 7.13 | 1717.0 | −41.6 | −64.1 | 45.2 | 0.0028 | 3632 |
| | G_front_middle | 6.41 | 1377.1 | −40 | 10.9 | 40.4 | 0.0076 | 2367 |
| Right | S_precentral-inf-part | 4.06 | 1303.4 | 34.2 | 5.6 | 33.5 | 0.0092 | 2282 |
| | S_subparietal | 3.52 | 609.5 | 7.8 | −46.4 | 42.2 | 0.0494 | 1619 |

For each contrast displayed in Fig. 2, and each cluster surviving $p_{FWE}$ < 0.05 (corrected at cluster level by permutation methods with cluster forming threshold $p$ < 0.01), the table reports: the cluster label (as defined by the anatomical labels from the Destrieux Atlas), the maximum −log10($p$) value in the cluster (Max), the cluster surface area in mm² (size), the MNI coordinates of the maximally activated vertex within each cluster (MNI X, Y, Z), the cluster-wise $p$-value of each cluster (CWP), the number of vertices included in each cluster (NVtxs). No above threshold clusters were detected for the operation at any time window, nor for the operand and result at the early and middle delay.

only for the sample and result in isolation but also attempting to predict the result from the sample and vice versa, as illustrated in Fig. 4b. In none of the ROIs tested, cross-decoding between early sample-evoked activity patterns and later (pre-probe) result-evoked activity patterns remained significant across subjects after correction for multiple comparisons across ROIs. Full statistical results for all ROIs can be found in Supplementary Table 3. This means that, on average across the group, cross-decoding was absent in the regions which in the preceding representational similarity analysis, were found to reliably represent both the result and the sample numerosities.

We reasoned that the absence of significant cross-decoding performance could be due to several factors: Beyond a true absence of a shared representational space, it could reflect a limited signal-to-noise ratio and / or statistical power in the present paradigm with small amounts of trials/stimulus presentations. On the other hand, if representations were shared at some level of the hierarchy, our ability to detect this would, in addition, depend on the degree to which participants actually are successful in generating the correct result numerosity. Failing to do so would be expected to manifest in reduced accuracy in the comparison task with the probe stimulus appearing at the end of the delay. We therefore correlated, across subjects, the above-mentioned cross-decoding performance with the subjects' behavioral Weber fraction for comparing the result with the probe numerosity (see Fig. 4b). Among the parietal ROIs, MIP was the only region showing a significant (negative) correlation, corrected for multiple comparisons across all parietal and frontal subregions tested (Pearson $r$(15) = −0.8, $p_{FDR}$ = 0.0035). Full statistical results for all ROIs can be found in Supplementary Table 4. This finding, in addition to the fact that MIP was one of the regions representing significant information about both sample and result numerosity in the previously

described multiple regression RSA ROI analysis, supports the idea of a role of this region in supporting a shared representational space between the sample and result numerosities in the present task, identifying it as a candidate cortical location where the transformation from in- to outputs of the computation could take place. An equivalent analysis for frontal regions of interest did not reveal any significant correlations (neither at corrected nor uncorrected level; see Supplementary Table 4).

## Discussion

The present study exploited the enhanced signal-to-noise ratio and functional sensitivity afforded by ultra-high-field imaging to characterize brain representations of internally generated numerical contents beyond those representations that arise simply as a consequence of the sensory processing of a stimulus (which has been in the focus of multiple previous studies). Our specifically designed paradigm and analytic approach allowed us to disentangle activity patterns reflecting the result of an approximate numerical computation from those representing the sample numerosity that was visually presented to the subject, as well as the other associated aspects of the computation. The representation of the physically presented sample numerosity was most widespread and most easily detectable, involving the dorsal visual stream hierarchy in addition to lateral prefrontal and occipito-temporal regions, compatible with a range of previous studies that either found numerical information detectable by multivariate decoding or explicit topographic layouts of numerosity responses, in most of these regions[8,9,15,16]. During later parts of the trial delay period, result numerosity representations could be detected, most significantly in higher-level cortical regions such as the angular gyrus and lateral prefrontal cortex. To our knowledge, no such brain

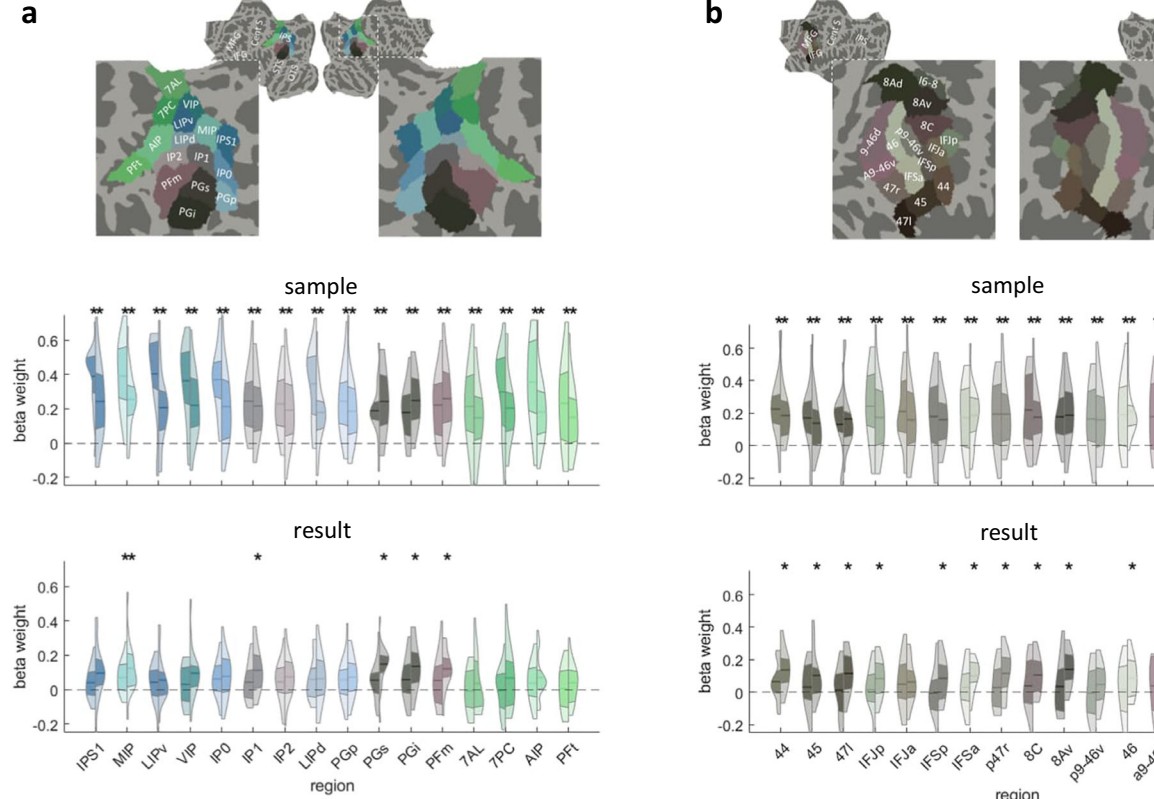

**Fig. 3 | Effects of sample and results numerosity in representational similarity analysis on regions of interest (ROIs).** Results for sample and results numerosity predictors in a multiple regression analysis on brain activity pattern dissimilarity are shown for subregions of the parietal (**A**) and lateral prefrontal (**B**) cortex. ROI delineation, colors, and labels are based on the Human Connectome Project (HCP-MMP1) parcellation[30] shown here on the flattened cortical average surface. For each ROI, violin plots show the distribution of beta estimates for sample and result predictors (*n* = 17 subjects). The central horizontal line indicates the mean, and the adjacent colored area (colors corresponding to the ROIs above) the limits of the first and third quartile, with the left part of the violin for each ROI corresponding to averages of the middle (4–8 s), and the right part to averages of the late (8–12 s) delay period. Star signs (*) indicate significantly larger responses than 0 in two-tailed *t*-tests after false discovery rate (FDR) correction for multiple comparisons ($p_{FDR} < 0.05$, across all 32 subregions and the two time periods). Source data are provided as a Source Data file.

representation of internally generated results of numerical computations has previously been reported, while one previous study using a slightly different paradigm involving subtraction or addition of small symbolic numbers and magneto-encephalography tried but did not succeed in detecting the representation of the internally generated outcomes[31].

The fact that result representations were most detectable in our study during the last four seconds of the delay period preceding probe appearance (though also see results for region MIP in the RSA ROI analysis in Fig. 3, and differential contrasts between middle and early delay for the result in Supplementary Figure 2), could be related to inherent constraints of the type of signals measured in fMRI rather than reflecting the actual time when results are available in the participant's mind. Indeed, given the temporal characteristics of the hemodynamic response, although the neuronal representation of the result could be available quickly after the operation instruction screen, the visually evoked BOLD response to the sample numerosity at that moment in time will still be strong and likely be overshadowing the internally computed result response. The use of long delays before probing the subjects with a comparison task was, therefore, a means to better separate the two types of responses (since the stimulus-evoked sample-related activity is expected to decay with time while the result has to be maintained for task purposes).

Given that our task involved internal maintenance of result representations and a following comparison task with a visually presented probe, a relevant question is in how far result representations could be contaminated with response selection processes on the one hand or reflect internal visualization of the outcome of the computation, potentially even in combination with non-numerical visual strategies. We think we can exclude a role of response selection confounds in affecting our findings since the different result numerosities were distinguishable before the appearance of the probe numerosity when participants could not yet predict the response to be made since probes differing by the same ratios towards the smaller vs. larger with respect to the result were used for each possible result. That participants were visualizing the outcome of the computation is possible, and we think, in general, this possibility is difficult to exclude also with other related approximate non-symbolic calculation tasks in the literature. If visualization occurs, we think that non-numerical visual strategies based on something else than the approximate number of discrete items, such as, for example, manipulating density or field area, would have limited viability since the field area within which the dots appeared (which was uncorrelated with numerosity by design) was always different between the sample and probe numerosity in our paradigm, thereby reducing correlations with density and their exploitability for performing the task. More generally, visual numerosity itself has been shown to be a salient visual property, for which perceptual sensitivity can insufficiently be explained by sensitivity to properties such as field area and density, which in theory jointly determine numerosity[32].

Among the regions found to represent result numerosities in the present study, the lateral prefrontal cortex is an area commonly found

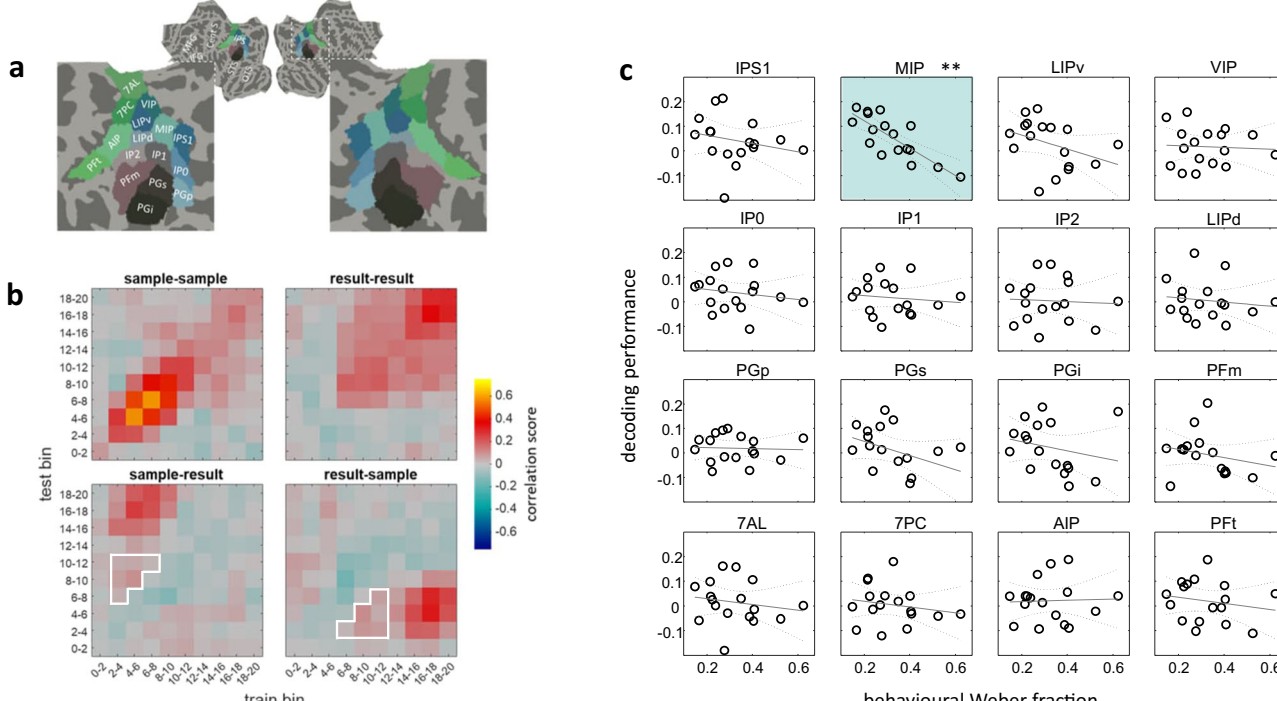

**Fig. 4 | Multivariate decoding analyses testing for shared representational spaces between stimulus-evoked and internally generated result representations. A** Visualization of all ROIs in the parietal lobe. ROI delineation, colors, and labels are based on the Human Connectome Project (HCP-MMP1) parcellation[30] shown here on the flattened cortical average surface. **B** Illustration of analysis: within each subregion, the performance of a multivariate decoder to predict the logarithm of numerosity for either the sample or result was tested after training on either the sample and results and the same as well as every other possible time bin (10 bins of the shown matrix). Decoding performance was measured by a correlation score (Fisher z transformed Pearson correlation between real and predicted labels). Average cross-decoding performance between sample and result for each subject was then assessed within the time window of interest outlined in white

corresponding to the periods where stimulus-evoked sample activity and internally generated result activity were hypothesized to be most prominent, respectively. The example data shown are the across-subject averages from the middle intra-parietal (MIP) ROI. **C** Pearson correlations across subjects ($n = 17$) between cross-decoding performance and behavioral Weber fractions for comparing the probe number with the internally generated result for all parietal subregions. Only in subregion MIP a significant negative correlation (Pearson $r(15) = -0.8$, $p_{FDR} = 0.0035$, ** indicating corrected significance across all parietal and frontal ROIs combined at $p_{FDR} < 0.01$) was found, indicating that better cross-decoding was associated with more precise behavioral numerical discrimination in the task (and thus, likely a more correct internal result representation). Source data are provided as a Source Data file.

to be recruited during numerical tasks in fMRI studies, most prominently in tasks involving arithmetic processing[4], and it also has been found to distinguish between individual numerical stimuli in both the visual and auditory modality by previous studies using fMRI pattern analyses which covered that region[9,12,33]. This region is also the one with the highest percentage of numerosity selective neurons in macaque monkeys[34]. Nevertheless, these neurons' responses arise later than the ones in the parietal cortex, and they are especially pronounced when monkeys have been trained to perform an explicit numerical match-to-sample task and reduced during non-numerical tasks, which is not the case for parietal responses[35,36]. This suggests that parts, but not all of, prefrontal numerical responses can be attributed to the more general encoding of task-relevant categories or working memory contents which is a hallmark of these regions also outside the numerical domain[37–39]. These aspects cannot be separated from the numerical contents in accounting for the representations of internally generated results, which we describe here for the first time, given the nature of the task which involved maintenance of and a comparison task carried out on the result. On the basis of previous fMRI work it seems, however, that mere working memory maintenance of perceived numerosity stimuli is not sufficient for their representations to be distinguished within these regions[19], although this may also reflect limitations of sensitivity of fMRI at lower field strengths.

The angular gyrus, which appears here as the other most prominent region representing internally generated results, is a location different from those known to show numerosity selective neurons in

monkeys and topographic numerosity maps in humans[15,34]. Indeed, in our paradigm, the angular gyrus ROIs were among the ones showing the least stimulus-evoked responses, contrasting with the other regions around the intra-parietal sulcus. The angular gyrus area is also known to be characterized by the absence of general visual topographic organization as found in more medial parts of the intra-parietal cortex, and it corresponds to the part of the parietal cortex having undergone most expansion in humans compared to macaques[40,41]. It is further considered a high-level integration zone with the greatest distance from sensory input regions which is at least partially overlapping with parts of the so-called "default mode network"[42,43]. Perhaps due to the latter proximity, the angular gyrus region is not among those regions generally observed to be overall activated during numerical as opposed to non-numerical tasks in fMRI studies[4]. Rather, when angular gyrus involvement was found in calculation tasks in humans, this has often been found to be specific to the operation, such as multiplication over subtraction, or been interpreted as related to different demands in verbal fact retrieval in the specific tasks compared[2,44,45]. An interpretation based on verbal retrieval appears unlikely for the findings here given the nature of our task and stimuli, and given the type of discrimination performed, which is between numbers rather than with respect to a control task with potentially differing retrieval demands, and is disentangled from the effect of the operation per se. Another situation where the angular gyrus has been found to be overall activated is when comparing numerical symbols relative to non-symbolic numerical stimuli[46–48], as well as for digits

compared to letters and scrambled characters during mere viewing[49]. The latter result has been interpreted as reflecting the symbols' conceptual identification, also compatible with findings showing preferential recruitment during conceptual (magnitude) as opposed to perceptual (color) judgments on numerals[50]. None of these studies, however, has gone beyond overall activation differences to demonstrate discrimination between symbols of different numerical magnitudes in this region.

Of note, before the introduction of functional imaging into the toolkit of cognitive neuroscience, the angular gyrus has already been established as a crucial substrate for mental calculation by neuropsychology[51–53]. Localized lesions to this area are known for yielding the Gerstmann syndrome of impairments, comprising acalculia, finger agnosia, agraphia, and right-left disorientation. Some authors[52] have proposed that the co-occurrence of these and further impairments in, e.g., mental rotation could have as a common denominator an impairment of visuospatial resources for manipulating mental images. Our results allow us to further qualify the functional role of this region, pinpointing its contribution in the context of calculation as representing individual quantities that are internally generated rather than other task-related components such as the operation per se. Interestingly, beyond the numerical domain, the angular gyrus was also found in other work to distinguish between newly learned multisensory objects, reflecting the learned concepts independently of their uni-modal sensory characteristics[54]. This suggests that this region's location on the top of multiple sensory processing hierarchies predisposes it to manipulate mental images of a rather abstract nature, which are detached from the specifics of the sensory inputs.

Beyond the results of the searchlight analysis, which reveal where numerical information is strongest across subjects when considering relatively locally restricted patterns, our complementary regions of interest analyses showed that representations of sample and results numerosities at an intermediate spatial scale are co-localized and relatively widespread, especially in the lateral prefrontal cortex. In the parietal cortex, co-localized information was detected in a few additional regions in the IPS that had not been observed in the searchlight analysis. This raises the question of whether and where the two different types of quantities are coded within a shared representational space. Indeed, our cross-decoding analysis between sample and result numerosities indicated that the only region where a shared code is likely based on our data was one region (MIP) on the medial wall of the IPS, where cross-decoding was not sufficiently significant across subjects to survive multiple comparison correction, but strongly significantly correlated with the behavioral precision for comparing the result with the probe numerosity presented at the end of the trial. This finding suggests that in this region, representations are indeed shared between sample and result, but that detectability with our methods was limited by the extent to which subjects computed the correct result.

The precise region where the above-mentioned result was obtained appears to be close to the location where topographic numerosity responses were first described[13]. The region is named MIP in the parcellation scheme applied here, reflecting the hypothesis of a possible correspondence with the medial intra-parietal area (or so-called parietal reach region) in macaques[55]. On the other hand, we note that this ROI, to a large extent, overlaps with field map IPS2 according to a different probabilistic atlas based on visual topography[56]. This indicates that it could also correspond to (part of) the equivalent of the lateral intraparietal area as suggested by others[57]. While disambiguating the precise correspondence is beyond the scope of our study, of interest for our interpretation here is the fact that both MIP and LIP are regions that are involved in sensorimotor integration with the effector corresponding either to the arm or the eye[58], as part of which these regions are thought to

implement coordinate transformations[59] to compute the direction and / or amplitude of movements. It has been hypothesized that the same neural machinery might be involved in numerical computations during mental arithmetic[60]. Previous studies have found that different types of numerical operations (such as addition versus subtraction or subtraction versus multiplication) performed on non-symbolic or symbolic numerical stimuli evoke dissociable activity patterns in the human LIP equivalent[61,62], but what was lacking was a demonstration of the representations of numerical outcomes actually being generated within these regions. In this context, our results provide a crucial missing link in support of the important role of sensory-motor regions in numerical computations. It is noteworthy that a shared representational space in our study was not observed in further regions beyond the ones on the medial wall of the IPS, in spite of the fact that the angular gyrus ROIs and many lateral prefrontal ROIs contained information on both the sample and result numerosities. While we cannot rule out limitations in sensitivity underlying this absence of significant generalization of decoding, it is possible that these higher-level regions, due to being more detached from sensory inputs, maintain sample and result representations in separate subspaces to maximize efficient coding of the task-relevant information and avoid mutual overwriting of the in- and outputs of the computation.

In sum, exploiting the enhanced signal-to-noise ratio provided by ultra-high-field fMRI in combination with an original paradigm, our study reveals a representation of internally generated quantities during mental computations, most strongly in higher-level regions on top of the sensory processing hierarchies and different from those known to be most strongly distinguish perceived quantities. Furthermore, regions along the medial wall of the IPS were the only ones to demonstrate some degree of shared representational space between the in- and outputs of the mental computation, identifying them as possible candidates for the initial generation of numerical outcomes, which should be further confirmed by methods with better temporal resolution. While the generalization of decoding performance suggests that perceptual and internally generated contents share some aspects of their cortical representation in these regions, another remaining question beyond the capacity of the imaging methods used here concerns how precisely these representations co-organize along the cortex and/or across cortical depth. It would be an interesting challenge for further studies to explore with ultra-high spatial resolution (potentially at even higher field strength) whether sample and result numerosities differentially recruit different cortical laminae, similar to what has been found for bottom-up and top-down generated representations in early sensory cortices[63,64]. The results reported here were obtained during mental manipulation of non-symbolic numerical quantity in an approximate calculation task. In how far some of the regions observed here would also be relevant to encode results of precise symbolic calculation, and what would be the role of different strategies employed (e.g., explicit manipulation of quantity versus retrieval from verbal memory) in the precise form that such result representations can take, will be important remaining questions for future studies.

## Methods

### Participants

18 healthy volunteers (25.9 ± 6.9 years old, of which 9 were male and 9 were female, according to self-report) were included in the study. One male participant was excluded due to not properly understanding and following task instructions while in the scanner (pressing the response button erratically during the sample or probe period). The study had been approved by the regional ethical committee (Comité de protection des personnes [CPP] Ile de France VII, Hôpital de Bicêtre, No. 15-007), and all participants gave written informed consent before participating.

## Data acquisition

Functional images were acquired on a SIEMENS MAGNETOM 7 T scanner with whole-body gradient and a 1Tx-32Rx head coil (Nova Medical, Wilmington, MA, USA) as T2*-weighted fat-saturation echo-planar image (EPI) volumes with 1.5 mm isotropic resolution using a multi-band sequence[65] (https://www.cmrr.umn.edu/multiband/, multi-band [MB] = 2, GRAPPA acceleration with [IPAT] = 2, partial Fourier [PF] = 7/8, matrix = 130 × 130, repetition time [TR] = 2 s, echo time [TE] = 22 ms, echo spacing [ES] = 0.64 ms, flip angle [FA] = 68°, bandwidth [BW] = 1832Hz/Px, phase-encode direction anterior » posterior). Calibration preparation used Gradient Recalled Echo (GRE) data. 68 oblique slices covering the occipito-temporal, parietal, and frontal cortex were obtained in ascending interleaved order. Before the experimental runs, two single volumes were acquired with the parameters listed above but with opposite phase encode directions to be used for distortion correction in the later analysis (see "Image processing and data analysis"). T1-weighted anatomical images were acquired at 1 mm isotropic resolution using an MP2RAGE sequence (GRAPPA acceleration with [IPAT] = 3, partial Fourier [PF] = 6/8, matrix = 256 × 256, repetition time [TR] = 5 s, echo time [TE] = 2.84 ms, time of inversion [TI] 1/2 = 800/2700 ms, flip angle [FA] 1/2 = 4°/5°, bandwidth [BW] = 240 Hz/px).

## Stimuli and procedure

Visual stimuli were back-projected onto a translucent screen at the end of the scanner bore and viewed through a mirror attached to the head coil. Participants held two response buttons in their left and right hands. Each trial started with the fixation cross turning green and a 400 ms presentation of a cloud of simultaneously presented visual items (sample: 6, 12, 24, or 48 dots) from which participants had to extract their numerosity. This was followed by a visually presented symbolic cue, appearing 2 s after the initial sample display for 400 ms, instructing participants about the operation to perform on the sample numerosity (multiplication/division by two/four). After that, participants had to maintain the internally generated result of the operation in memory over a prolonged delay period until another set of dots (probe numerosity) appeared on the screen. Participants were asked to perform a comparative judgment on the probe with respect to their internally generated result by pressing the left button when the probe appeared smaller and the right button when the probe appeared larger. Probe numerosities varied from the correct result of each given operation by one of 8 ratios (0.5, 0.667, 0.8, 0.909, 1.1, 1.25, 1.5, and 2), allowing us to estimate participants' just noticeable difference (JND) on a log scale (Weber fraction). Given that all smaller and larger probe ratios appeared with equal probability for each possible result, the detected result representation cannot be systematically affected by response selection confounds before probe appearance. The SOA between sample and probe number was 12 s on the majority of trials (standard trials), and was unpredictably shortened (randomly drawn from a uniform distribution ranging from 5.4 s to 9.6 s) on 20% of trials (catch trials). These catch trials were included to encourage subjects to perform the computation immediately and maintain the result representation ready for a potential comparison throughout the delay period.

The clouds of dots presented black items on a mid-gray background, which could appear within a circular field area of either 6 or 8 visual degrees diameter for all numerosities. Sample and probe numerosity were always presented in different field areas to minimize the possibility of using dot density to perform the task. The size of the dots was adjusted so that the total luminance was approximately equated across numerosities. Individual dot sizes were picked from a uniform distribution whose mean varied with the numerosity. Each half of the distribution overlapped with a consecutive numerosity.

Calculation paradigms inevitably yield some form of correlation between the results and either the operation or the sample. In our case,

the 4 numerosities, 2 operations, and 2 operands were used to create 10 trial types/conditions (Fig. 1), ensuring that the operation was equally correlated with the sample and result, respectively. Two standard trials of each of these 10 conditions appeared within a given experimental run, which in addition contained 4 catch trials (not included in the analysis). All subjects completed between 6 and 8 runs (of ~8 min duration each).

## Image processing and data analysis

EPI images were motion-corrected and co-registered to the first single-band reference image using statistical parametric mapping software (SPM12, https://www.fil.ion.ucl.ac.uk/spm/software/-spm12/). The single-band reference images of the two initial volumes acquired with opposite phase encode directions served to estimate a set of field coefficients using top-up in FSL 5.0 (https://fsl.fmrib.ox.ac.uk/fsl/fslwiki/FSL), which was subsequently used to apply distortion correction (apply_topup) to all EPI images. Cortical surface reconstruction of the anatomical image and boundary-based registration of single band reference images to each subject's cortical surface, as well as a minimal amount of surface-constrained smoothing (FWHM = 2 mm) for noise reduction, were performed in Freesurfer 6.0 (https://surfer.nmr.mgh.harvard.edu/). The preprocessed EPI images (in subjects' native space) were entered into a general linear model, using a finite impulse response (FIR) set with 10 bins of 2 s (=1 TR) width to model responses to standard trials separately for the 10 experimental conditions, as well as an additional condition corresponding to all catch trials. The six motion parameters were included in the GLM as a covariate of no interest. An AR(1) model was used to account for serial auto-correlation, and low-frequency signal drifts was removed by a high-pass filter with a cutoff of 244 s.

Pattern analysis testing for separable representations of the different components of the calculation task was performed using a representational similarity analysis approach in CoSMoMVPA (http://www.cosmomvpa.org/)[66], with as input a single set of FIR estimates per condition and subject, obtained from a GLM concatenating the runs within-subject, also see[16,67]. In addition to the movement parameters, this GLM contained a separate constant for each run to account for offsets between runs. Searchlight analysis in each individual subject's volume space was performed for each of the 10-time points (FIR bins) after the sample, using a spherical ROI with a radius of 3 voxels. Within the searchlight sphere, voxel-wise scaling was applied by subtracting the mean across conditions, and neural representational distance matrices (RDMs) were created by computing the correlation distance (1−the Pearson correlation across voxels) between activity patterns associated with all possible pairs of the 10 conditions. The neural RDMs were then entered in a multiple regression with four predictor RDMs encoding the differences between the 10 conditions in terms of the sample and probe numerosities (log scale), operations, and operands. All distance matrices were z-transformed before estimating the regression coefficients. These analyses yielded as output a set of beta estimate images (corresponding to the effects explained by the 4 predictor RDMs on the neural RDMs for all searchlight center locations) for each time bin.

Beta images of individual subjects were then projected onto their cortical surface for group analyses (using Freesurfer), performed on each one of three-time windows of interest corresponding to early (average of bin 1 and 2), middle (average of bin 3 and 4) and late (average of bin 5 and 6) parts of the trial delay period (all before the probe numerosity appeared on the screen). Equivalent pattern analyses were also performed at the scale of multiple subregions of the parietal and lateral prefrontal cortex, as defined by a multimodal parcellation scheme in standardized cortical surface space[30]. The analysis focused on a set of 16 subregions, each within the parietal and prefrontal cortex (see Fig. 3 for the different regions' locations). More specifically, the included subregions' labels were: PFt, AIP, 7PC, 7AL,

PFm, PGi, PGs, PGp, LIPd, IP2, IP1, IP0, VIP, LIPv, MIP, IPS1 for parietal cortex, and 8Ad, i6-8, 9-46d, a9-46v, 46, p9-46v, 8Av, 8 C, p47r, IFSa, IFSp, IFJa, IFJp, 47l, 45, 44 for prefrontal cortex. Regional labels in normalized cortical surface space were projected back into each individual subject's volume space to create regions of interest. Preliminary analyses kept left and right hemisphere ROIs separate, which were then merged after observing no significant interactions between any effects for the predictors of interest and hemisphere in an ANOVA.

Multivariate decoding analysis testing for cross-prediction between sample and result numerosities was performed on the same set of regions of interest as mentioned above in scikit-learn (https://scikit-learn.org/stable/)[68] using linear support vector regression (SVR) with regularization parameter $C = 1$. This analysis was performed on run-wise FIR beta estimates of the 10 conditions in each subject, implementing a cross-validation loop that left patterns of one run out for testing in each cycle and training on the remaining ones in an equivalent way for all the comparisons mentioned below. The decoder was trained on data that were voxel-wise z-scored across conditions, with continuous labels reflecting the z-scored logarithm of number (either of the sample or the result). The prediction performance on the left-out test data was evaluated by a correlation score (Fisher-z transformed Pearson correlation coefficient between real and predicted labels). This procedure was applied either within the sample or result numerosities, but most importantly, to predict the sample from the result and vice versa, where in addition, the train and test data could come from all possible combinations of the 10 FIR bins, yielding a $10 \times 10$ matrix of decoding generalization performance. Since analyses using a flexible, parameter-rich FIR model tend to yield noisy estimates for individual bins, we focused on averages of multiple cells in these matrices, which reflected our prediction that in case of a shared representational space, cross-decoding of numerosity should be observed between time points reflecting for the sample the typical peak of the stimulus-evoked response, and for the result later periods during the delay where the stimulus-evoked response was expected to have diminished and mostly the memorized result representation to remain. In detail, for the sample, this comprised bins 2–4 (corresponding to 2–8 s post-stimulus onset), and for the result, bins 4–6 (corresponding to 6–12 s post-sample onset), with the additional constraint that the combinations of bins retained needed to be non-overlapping for sample and result and separated by at least 1 bin in between them (see Fig. 4 for illustration).

## Statistical analysis

Group analysis of the multivariate searchlight representational similarity analysis in surface space was performed using two-tailed t-tests against 0 across the subject-wise beta estimates for each predictor. Correction for multiple comparisons was performed at the cluster level using permutation methods as implemented in Freesurfer (5000 permutations, $p_{FWE} < 0.05$, with a vertex-wise cluster forming threshold of $p < 0.01$). Equivalent analyses with two-tailed t-tests against 0 across subjects were conducted in the ROI analysis on the subject-wise beta averages for each ROI. Multiple comparison correction was performed using the Benjamini & Hochberg method for the correction of false discovery rate (FDR) across all 32 subregions (frontal and parietal combined) and both time windows (middle and late combined).

For the ROI cross-decoding analysis, the values entering two-tailed t-tests against 0 were the subject-wise Fisher-z-transformed decoding scores for the time window of interest defined in the section above. Multiple comparison correction was based on FDR across all 32 subregions (frontal and parietal combined). Correlation analysis across subjects between subject-wise decoding scores and measures of behavioral performance (just noticeable difference/Weber fraction) was performed using Pearson correlation and the associated significance test, with FDR-based multiple comparison correction across all subregions.

## Reporting summary

Further information on research design is available in the Nature Portfolio Reporting Summary linked to this article.

## Data availability

The functional imaging data (individual subjects' first-level fMRI models, anatomical images, and ROI images) generated as part of this study have been deposited in an Open Science Framework database (https://osf.io/2tnrz/). Source data plotted in the figures are provided with this paper. Source data are provided with this paper.

## Code availability

The analysis code used in this study (representational similarity and multivariate decoding analysis) is available via the Open Science Framework (https://osf.io/djftw/).

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

## Acknowledgements
This work was funded by the French National Research Agency via grant No ANR-14-CE13-0020-01 to E.E. The 7 T MRI device at Neurospin has received support from the Leducq Foundation (ERPT Equipment Program).

## Author contributions
Conceptualization: S.C. and E.E.; methodology: E.E. and A.V.; software: S.C. and E.E.; validation: E.E.; formal analysis: S.C. and E.E.; investigation: S.C. and E.E.; data curation: S.C., A.V. and E.E.; writing—original draft: E.E.; writing, review and editing: A.V. and E.E.; visualization: E.E.; supervision: E.E.; project administration: E.E; funding acquisition: E.E.

## Competing interests
The authors declare no competing interests.
