## [Peer Review File · Nature Communications]

Human brain representations of internally generated outcomes of approximate calculation revealed by ultra-high-field brain imagingREVIEWER COMMENTS

Reviewer #1 (Remarks to the Author):

Overall this is an exciting study which provides the first evidence (that I know of) suggesting any neural response that might be associated with internally-generated calculation results. It is well written, clear and cautious about the implications of its results. The analysis is technically sophisticated. At the same time, the background introduction and discussion are both detailed, up-to-date and accessibly written. All of this makes for enjoyable reading.

However, a few aspects later in the results are much less convincing, and in my view don't convincingly support parts of the story, particularly regarding cross-decoding between sample and result, and the correlation with behaviour. While these are presented mostly carefully in the results and discussion, the abstract presents them with a certainty that is not supported by the results.

Finally, it is not at all clear whether the decoded result reflects specifically numerical aspects of the result. The authors are pretty clear about what might underlie result decoding, and do not just attribute it to an internally-generated number concept. So I still find this an exciting step forward in understanding the processes underlying numerical cognition, even where it is not yet clear what form this result-specific response takes.

Major points:

First, I am not at all convinced that the MIP ROI allows cross-decoding between the sample and result numerosity. The authors are clear in the results that: "Among the multiple regions of interest within parietal cortex, we found that the medial intraparietal area (MIP) region of interest located on the superior wall of the IPS was the only region allowing for some amount of cross decoding between early sample-evoked activity patterns and later (pre-probe) result-evoked activity patterns ($t(16) = 2.57$, $p=0.02$, uncorrected, although not surviving multiple comparison correction across multiple regions, $pFDR = 0.33$)." This final corrected probability is the important statistic here, and there is clearly no significant effect after this correction for multiple comparisons. While the authors are careful about this in the results and discussion, the abstract simply states "subregions on the medial wall of the intraparietal sulcus allowed for cross-decoding of the sample and result numbers". A lot of readers do not get beyond an abstract and take its claims at face value. I guess the authors are just trying to be brief in the abstract, but I find this phrase unacceptably misleading. I completely agree with the final sentence "resulting outputs maintained for task purposes in higher-level regions in a format possibly detached from sensory evoked representations".

Second, I am not at all convinced that the same ROI shows a meaningful correlation with behavioural performance. This correlation survived correction for multiple comparisons, though here the comparisons included ignore those made in the frontal lobe. But more concerning, the correlation is negative: lower cross-decoding performance comes with higher behavioural precision. This does not then support the conclusion in the abstract that “cross-decoding of the sample and result numbers... reflected the behavioral precision of the result representation, indicative of a shared representational space between perceived and internally computed quantities.” Did the authors expect a negative correlation? If so, they should explain why, because I do not see it. Instead, if there is a link here between neural activity and behaviour it is something along the lines of ‘more similar representations lead to the participant giving a response to the sample numerosity rather than the result numerosity’ (my words, not the authors). That would seem highly speculative. In the abstract, results and discussion, this result is presented as being very clear.

I also do not agree with the authors’ choice to describe the result numerosity as ‘number’ rather than ‘numerosity’. To an extent, I see the motivation that the sample is a numerosity and the operand is a symbolic number so maybe the result is neither or both. We are not completely sure what is happening in the participant’s mind here. But I find this hard to swallow because the probe display is non-symbolic, numerosity. Using the term ‘number’ for the result (rather than ‘numerosity’) implies a generalised number representation, and I expect many readers will understand the claims as such. As the authors know, responses to specific symbolic numbers or generalised number concepts are much less decodable than responses to specific numerosities. Please be much more careful with the term ‘number’ throughout: say numerosity when that’s what you mean (including the abstract), and explain the nuances of the concept of ‘number’ in interpreting the results.

I think it should also be explained in the main manuscript (not hidden in the methods) if there is really NO correlation between the sample and result. As I understand it, larger samples generally come with larger results. So the sample and results predictors do not seem to be truly independent and the linear model assigning parts of the response to sample and result may assign some of the response variance to the wrong predictor. I may be wrong about this, but if so this should be explained. Even if I am correct, I do not see this as a major drawback: it is notoriously complex to separate correlated features in numerical tasks. Instead, this issue needs to be explained in more detail and (if appropriate) applied to the interpretation of the results.

Related, the authors make a lot of the fact that the result response is limited to the late time period. In my understanding of the task timing, the result is available in the mid-period, within 2 seconds (1 TR) of the sample. Then why is the response to this limited to the late period?

A response limited to the late period may have other implications concerning the participant’s strategies in performing the task, and aspects of these potential strategies may allow decoding of the result. For example, response selection: is the response the participant will give completely uncorrelated with the

calculation result? Is it likely that the participant is verbalising or visualising the result during the delay period?

Regardless of these last points, the authors are pretty clear about what might underlie result decoding, and do not just attribute it to an internally-generated number concept. For example, they explain that “This suggests that parts, but not all of, prefrontal numerical responses can be attributed to more general encoding of task-relevant categories or working memory contents which is a hallmark of these regions also outside the numerical domain. These aspects cannot be separated from the numerical contents in accounting for the representations of internally generated result numbers which we describe here for the first time, given the nature of the task which involved maintenance of and a comparison task carried out on the result.” So I still find this an exciting step forward in understanding the processes underlying numerical cognition, even where it is not yet clear what form this result-specific response takes. It would be great if the authors could make the abstract and perhaps the title clearer about this important uncertainty.

Reviewer #2 (Remarks to the Author):

The paper by Czajko et al. entitled “Human brain representations of internally computed quantities revealed by ultra-high-field brain imaging” presents the results of a 7T fMRI study which show that physically presented sample numbers can be decoded from the dorsal visual pathway whereas numbers of internally generated results are detected in the angular gyrus and the lateral prefrontal cortex. Furthermore, cross-decoding of the sample and the result number are revealed in the medial wall of the intraparietal sulcus (reflecting also the behavioral precision of the results representation). In my view, this is a very well-designed study that provides new results on the neural representation of internally generated numbers. The results are of high scientific interest and the methods state-of-the-art. I have only a few minor comments that the authors should consider:

Methods:

Could the authors explain if the concatenation of runs in the GLM is a mere concatenation, and if the model contains a run-constant to model potential offsets between runs and provide a reference where the same procedure has been previously applied (Line 436)?

The authors should motivate why they used these particular toolboxes for decoding. It would certainly be nice and clear if only one toolbox was used, for example ‘The Decoding Toolbox’. The authors should therefore consider (re)-running the analyses again with this program to standardize/unify their results.

Significance testing for the cross-decoding analysis could be added.

Maybe you could specify your FDR correction method in a bit more detail, e.g. “Multiple comparison correction was based on FDR across subregions within lobe”, by motivating why it is done within lobe and providing numbers of the amount of regions and cut-off significance values as well as p-values where possible.

Related to this, I am not sure if it is helpful to report in line 179 uncorrected $p < 0.05$ significance – maybe see also comment on FDR correction, it would help to see where FDR correction is doing the cut-off to evaluate this.

What does “multiple regression” (line 652) refer to? Please describe this in more detail

With regards to Tables and Figures:

Organization of Table1: Typically the anatomical regions are written on the left.

The authors use different types of surface renderings in Figure 1 and for the presentation of the ROIs. Would it be possible to use the same display to make the findings more accessible? It might be much to ask for – and it might be a bit tricky to do such a visualization, but maybe one could also overlay the borders of the surface parcellation (from Figure 3) with the results of the searchlight analysis in Figure 2. Alternatively, one could try to plot the results presented in Figure 3 into the same kind of surface rendering as presented in Figure 2 (by dropping the error bars, and coloring each ROI according the beta weight value.

Figure 1: the table might be very small in the final print. Was the ITI fixed to 7.2 sec? Why?

The correspondence of the RSA lines and columns could be made more clear

Figure 2: The authors could additionally report contrasts of e.g. early > late or early > (mid & late) etc. (maybe in the supplement)

Figure 3: All x-axis for the beta weights should be scaled in the same way, as otherwise one is tempted to compare the effect sizes between the sample and results, which are not scaled in the same way.

Figure 4A: I do not understand from the figure nor from the text how the visualization of the ROIs in Figure 4A relates to the plots below. Three of the scale bars can be removed and the four matrices can be aligned so that the labelling of the x- and y-dimension is only provided once. These types of plots are typically plotted in a way so that the diagonal is from the bottom left to the top right (See EEG-decoding literature), and I would consider it helpful to stick to this way of visualization. It remains unclear for which ROI this is plotted?

Figure 4B: Plots could be made more efficient by dropping redundant axis labelling.

There are 5 figures but only 4 figure legends. Figure 5 (I assume the supplementary one): As for Figure 3, all axis should be plotted on the same scaling.

Reviewer #3 (Remarks to the Author):

Summary

In this study Czajko, Vignaud and Eger investigate the neural representation of internally generated outcomes of approximate (i.e. non-symbolic) multiplication and division problems by means of high-field neuroimaging. After having been presented with a set of dots (sample) participants were prompted with a multiplier/divisor in symbolic format (either 2 or 4, operand). After the delay period during which they were asked to maintain the result in working memory, participants were probed with a comparison numerosity (probe) that they had to compare to the maintained outcome.

Using a searchlight algorithm for detecting BOLD signatures of operands, operation and outcome in a 'time-resolved' (i.e. TR-wise) fashion, the authors detected marked effects of the sample numerosity in bilateral parietal areas along the intraparietal sulcus and "lateral prefrontal and occipito-temporal brain regions". In time bins immediately preceding the probe, signatures of the outcome were observed in left middle frontal gyrus, the right inferior precentral sulcus, the left angular gyrus and the right medial parietal lobe. Additional analyses revealed signatures of the result in superior wall of the IPS (regions IP1 and MIP) in the same time window. Common coding of (early) sample and (late) result numerosity was observed in MIP. The decoding accuracy in this analysis correlated with the behaviorally observed JND across participants.

These results are interpreted as evidence for a transformation of sample into result quantities in dorsal stream sensory-motor integration regions. The generated results would further be maintained for task purposes in higher-level regions in a format possibly detached from sensory evoked representations, involving angular gyrus, middle frontal gyrus and precentral sulcus.

Evaluation

This study tackles an important question in numerical cognition. What cortical circuits are involved in generating and maintaining the outcome of arithmetic problems and to what extent do these circuits overlap with those involved in decoding the perceptual input numerosity? This question – albeit being central to much of mathematical cognition – has remained unanswered so far. Most work in this domain has been devoted to the deciphering of the neural codes underlying perceived quantities. Hence, the results present an important step towards a comprehensive understanding of the processing steps that are involved in solving arithmetic problems. The study is expertly designed and analyzed. The manuscript is overall well written but some formulations may merit the authors' attention. Overall, I do not see any reasons that would prevent this study from ultimately being published. I think it would be well received in the community. The authors may find the following comments helpful in further improving the manuscript.

Comments

The authors chose an intermediate step by investigating non-symbolic (approximate) arithmetic - a choice presumably driven by the difficulty of decoding symbolic quantities. It may nevertheless be helpful to inform the reader why the current design – in particular the non-symbolic format – was chosen over more common symbolic arithmetic problems. In this context, a distinction that is often found in the literature may be discussed: approximate versus exact calculation.

What implications do the current results have for symbolic calculations?

The non-symbolic format may afford visual strategies to solve the given problem. That is, participants may imagine the individual dots split in two or four for multiplication or to merge for division. One may hence argue that no numerical representation is required to solve the task because participants may rely on non-numerical features. While I am not arguing that this is necessarily the strategy that participants adopted because numerosity is a salient stimulus dimension that is readily used by participants, it may still be interesting to discuss this possibility.

Did authors adopt a more classical analysis approach (i.e. convolving the signal with a canonical HRF) and tried to identify regions that show numerosity-specificity or even place coding as shown for single parietal and prefrontal neurons in NHPs? Plotting the time-course in ROIs with significant decoding would be informative, too.

My copy of the manuscript contained a figure 5 for which I found no caption and that was not referenced in the text body.

Authors report overall accuracies of ~75%. It would be informative to report the accuracies as a function of ratio that allowed them to estimate JNDs. Also, did authors analyse the effect of ratio on BOLD?

When inspecting figure 4, I was wondering how authors would interpret the cross-decoding between early sample and late result that appears as 'red blob' next to (sample-result) and below (result-sample) the highlighted time window that authors focus on?

A second question that jumps to my mind is why this later time window is the same where authors observe the highest accuracy in decoding the result numerosity (result-result).

Minor comments

Line 64: "humans has proceeded to characterizing the representations / neural codes of numerical information" Please provide a verbal conjunction instead of "/"

170/171: "Results showed that information *about* the sample number was widely *distributed* in parietal cortex (Figure 3 A),

181: Overall, these ROI analysis result*s*

251: This region* is also the one

Reviewer #1 (Remarks to the Author):

Overall this is an exciting study which provides the first evidence (that I know of) suggesting any neural response that might be associated with internally-generated calculation results. It is well written, clear and cautious about the implications of its results. The analysis is technically sophisticated. At the same time, the background introduction and discussion are both detailed, up-to-date and accessibly written. All of this makes for enjoyable reading.

However, a few aspects later in the results are much less convincing, and in my view don't convincingly support parts of the story, particularly regarding cross-decoding between sample and result, and the correlation with behaviour. While these are presented mostly carefully in the results and discussion, the abstract presents them with a certainty that is not supported by the results.

Finally, it is not at all clear whether the decoded result reflects specifically numerical aspects of the result. The authors are pretty clear about what might underlie result decoding, and do not just attribute it to an internally-generated number concept. So I still find this an exciting step forward in understanding the processes underlying numerical cognition, even where it is not yet clear what form this result-specific response takes.

We thank the reviewer for the very positive general evaluation and present below our point-by-point response which should also clarify a few misunderstandings regarding the aspects judged as "less convincing".

Major points:

First, the I am not at all convinced that the MIP ROI allows cross-decoding between the sample and result numerosity. The authors are clear in the results that: "Among the multiple regions of interest within parietal cortex, we found that the medial intraparietal area (MIP) region of interest located on the superior wall of the IPS was the only region allowing for some amount of cross decoding between early sample-evoked activity patterns and later (pre-probe) result-evoked activity patterns ($t(16) = 2.57$, $p=0.02$, uncorrected, although not surviving multiple comparison correction across multiple regions, $pFDR = 0.33$)." This final corrected probability is the important statistic here, and there is clearly no significant effect after this correction for multiple comparisons.

We agree that the corrected statistics is the most important result to retain here, and therefore have changed the wording (on page 7): "In none of the ROIs tested, cross-decoding between early sample-evoked activity patterns and later (pre-probe) result-evoked activity patterns remained significant across subjects after correction for multiple comparisons across ROIs. Full statistical results for all ROIs can be found in Supplementary Table 3".

While the authors are careful about this in the results and discussion, the abstract simply states "subregions on the medial wall of the intraparietal sulcus allowed for cross-decoding of the sample and result numbers". A lot of readers do not get beyond an abstract and take its claims at face value. I guess the authors are just trying to be brief in the abstract, but I find this phrase unacceptably misleading. I completely agree with the final sentence "resulting outputs maintained for task purposes in higher-level regions in a format possibly detached from sensory evoked representations".

We have reformulated the phrase in the abstract to: "Behavioral precision in the task was related to cross-decoding performance between sample and result representations in medial IPS regions."

Second, I am not at all convinced that the same ROI shows a meaningful correlation with behavioural performance. This correlation survived correction for multiple comparisons, though here the comparisons included ignore those made in the frontal lobe.

We have repeated these analyses with multiple comparison correction across all ROIs including the frontal ones (32 regions in total), and the effect in MIP is still significant (see results page 7 and Figure 4).

But more concerning, the correlation is negative: lower cross-decoding performance comes with higher behavioural precision. This does not then support the conclusion in the abstract that “cross-decoding of the sample and result numbers... reflected the behavioral precision of the result representation, indicative of a shared representational space between perceived and internally computed quantities.” Did the authors expect a negative correlation? If so, they should explain why, because I do not see it.

We did not expect a negative correlation between precision and decoding performance. We apologize for the error in labelling the X axis which was referring to “behavioral precision” in the previous version. What is plotted here is the behavioral just-noticeable difference / Weber fraction (larger values = lower precision) therefore the relation with precision is actually positive. The label has been changed to “behavioral Weber fraction”.

Instead, if there is a link here between neural activity and behaviour it is something along the lines of ‘more similar representations lead to the participant giving a response to the sample numerosity rather than the result numerosity’ (my words, not the authors). That would seem highly speculative. In the abstract, results and discussion, this result is presented as being very clear.

No, more similar representations (better cross-decoding between sample and result) cannot arise from participants only maintaining the sample numerosity and potentially responding to that. If a subject would only represent the sample, the labels of result numerosities would not be predictable from the sample and vice versa (since sample and result numerosities are anti-correlated in our design, see Figure 1 top and bottom right), and behavioral performance for comparison with the probe should be bad. As a side note, we are not decoding “response” related activity here either, since during the delay period the subject cannot yet predict the required response because the probe numerosity can be either smaller or larger than the result, with equal probability).

By clarifying the presence of the labeling error and now correcting the statistical significance for all ROIs including the frontal ones, we hope to have convinced that this effect is more meaningful than it may have seemed in the previous version.

I also do not agree with the authors’ choice to describe the result numerosity as ‘number’ rather than ‘numerosity’. To an extent, I see the motivation that the sample is a numerosity and the operand is a symbolic number so maybe the result is neither or both. We are not completely sure what is happening in the participant’s mind here. But I find this hard to swallow because the probe display is non-symbolic, numerosity. Using the term ‘number’ for the result (rather than ‘numerosity’) implies a generalised number representation, and I expect many readers will understand the claims as such. As the authors know, responses to specific symbolic numbers or generalised number concepts are much less decodable than responses to specific numerosities. Please be much more careful with the term ‘number’ throughout: say numerosity when that’s what

you mean (including the abstract), and explain the nuances of the concept of 'number' in interpreting the results.

We did not mean to imply when using the term "number" that this necessarily refers to an abstract format (it is a "number" of dots), but we understand the concern and are using now "numerosity" whenever appropriate when referring to the internally computed results.

I think it should also be explained in the main manuscript (not hidden in the methods) if there is really NO correlation between the sample and result. As I understand it, larger samples generally come with larger results.

No, larger samples do not come with larger results due to the way the operations were assigned to conditions. Please see Figure 1 A, right, for the actual sample and result numerosities used across the 10 conditions, and 1 B, for the correlation of the predictors in the RSA analysis (which is -0.34 between sample and result). In addition to showing this in the figure, we now point it out in the figure legend.

So the sample and results predictors do not seem to be truly independent and the linear model assigning parts of the response to sample and result may assign some of the response variance to the wrong predictor. I may be wrong about this, but if so this should be explained. Even if I am correct, I do not see this as a major drawback: it is notoriously complex to separate correlated features in numerical tasks. Instead, this issue needs to be explained in more detail and (if appropriate) applied to the interpretation of the results.

It is indeed notoriously complex to separate correlated features in numerical tasks by design. This is why we have opted in the pattern analysis for an approach which includes the four different predictors at the same time into the multiple regression (where the resulting effects should then, on average across subjects, only reflect the part of the variance that each predictor can explain on top of all the others, and not the shared variance). We have added a few explanations related to this at the beginning of the fMRI result description (page 5). This analysis approach is equivalent to the one used in Castaldi et al., (2019), Elife, to separate numerical and non-numerical quantities, and the issue of correlation between predictors was already discussed there.

In the decoding analysis which is training and testing a multivariate (support vector) regression to predict the log-transformed numerical labels from multi-voxel brain activity patterns, effects explained by any other predictor than the one tested on cannot be removed. However, since this analysis is based on predicting the condition labels between sample and result as mentioned above, if anything, the negative correlation between sample and result would actually make it harder to find significant generalization between these two quantities (because if sample activity is driving result prediction, it would drive the predicted labels away from the correct ones of the result).

Related, the authors make a lot of the fact that the result response is limited to the late time period. In my understanding of the task timing, the result is available in the mid-period, within 2 seconds (1 TR) of the sample. Then why is the response to this limited to the late period?

When stating that we expect the result to be detectable during the late delay period, this is not referring to the actual task timing and the moment where the result should have been computed in the participants' mind, but to the time where we most expect to be able to detect it due to constraints of the methods. Due to the slow nature of the hemodynamic response, at the moment when the result is first computed, the strong activity related to sample and/or the operation will still

be present and likely mask the result related activity. The prolonged delay was chosen so that the activity evoked by these other factors would have at least to some degree returned towards baseline (not fully, this would have required even longer delays which were incompatible with acceptable trial numbers here) to improve the detectability of the maintained result representation, see e.g., Albers et al., 2013, *Curr Biol*, for a similar approach in the context of decoding a mentally rotated visual content. We have added a few sentences on these issues to the discussion (page 8). In addition to these purely methodological constraints, when the participants actually computed the result is somewhat beyond our control, we could only encourage them to do it immediately by instructing them to do so (and presenting some catch trials with a shorter delay period). Both of these factors may have contributed to the result being most detectable at the time points before the probe. As a side note though, in an additional analysis of differential contrasts between time periods (Supplementary Figure 2, Supplementary Table 2), a small cluster for the result could be detected in the angular gyrus already in the contrast of middle > early delay periods, and in MIP, the result representation survived correction for multiple comparisons during the middle delay (Figure 3).

A response limited to the late period may have other implications concerning the participant's strategies in performing the task, and aspects of these potential strategies may allow decoding of the result. For example, response selection: is the response the participant will give completely uncorrelated with the calculation result? Is it likely that the participant is verbalising or visualising the result during the delay period?

The response that participants have to give is indeed completely uncorrelated with the calculation result, this was achieved by presenting probe numerosities that could with equal probability be either smaller or larger than the result of the computation by equal ratios, for all possible results (see page 11 of the manuscript). We can therefore exclude that response selection confounds affected the result representation. It is, however, possible that participants were visualizing the result during the delay period, and such an explanation cannot be excluded by the present study, as we acknowledge now in the discussion (page 9). The degree to which these processes affected result representations in specific areas could only potentially be disentangled if future studies found ways to specifically induce one strategy over the other.

Regardless of these last points, the authors are pretty clear about what might underlie result decoding, and do not just attribute it to an internally-generated number concept. For example, they explain that "This suggests that parts, but not all of, prefrontal numerical responses can be attributed to more general encoding of task-relevant categories or working memory contents which is a hallmark of these regions also outside the numerical domain. These aspects cannot be separated from the numerical contents in accounting for the representations of internally generated result numbers which we describe here for the first time, given the nature of the task which involved maintenance of and a comparison task carried out on the result." So I still find this an exciting step forward in understanding the processes underlying numerical cognition, even where it is not yet clear what form this result-specific response takes. It would be great if the authors could make the abstract and perhaps the title clearer about this important uncertainty.

We have changed the abstract to refer to numerosity rather than number, but since the abstract is very short, we do not see how we can be clearer about this uncertainty in so little space. But any further suggestions would be welcome.

We have modified the title: Human brain representations of internally generated outcomes of approximate calculation revealed by ultra-high-field brain imaging

Reviewer #2 (Remarks to the Author):

The paper by Czajko et al. entitled “Human brain representations of internally computed quantities revealed by ultra-high-field brain imaging” presents the results of a 7T fMRI study which show that physically presented sample numbers can be decoded from the dorsal visual pathway whereas numbers of internally generated results are detected in the angular gyrus and the lateral prefrontal cortex. Furthermore, cross-decoding of the sample and the result number are revealed in the medial wall of the intraparietal sulcus (reflecting also the behavioral precision of the results representation). In my view, this is a very well-designed study that provides new results on the neural representation of internally generated numbers. The results are of high scientific interest and the methods state-of-the-art. I have only a few minor comments that the authors should consider:

We thank the reviewer for the positive general evaluation.

Methods:

Could the authors explain if the concatenation of runs in the GLM is a mere concatenation, and if the model contains a run-constant to model potential offsets between runs and provide a reference where the same procedure has been previously applied (Line 436)?

In the model with concatenated runs a constant was added for each run, in addition to the corresponding movement parameters, to account for the mentioned potential offsets between runs. We now explicitly mention this on page 14. Run concatenation is often performed in RSA studies to improve the reliability of the model estimation when designs contain many conditions and/or few trials per condition per run (for example see Kriegeskorte et al., 2008, *Front Systems Neurosci*), and we also have used it in previous work (Castaldi et al., 2019).

The authors should motivate why they used these particular toolboxes for decoding. It would certainly be nice and clear if only one toolbox was used, for example ‘The Decoding Toolbox’. The authors should therefore consider (re)-running the analyses again with this program to standardize/unify their results.

We used scikit learn (<https://scikit-learn.org/stable/>) for decoding analysis. These python libraries are a widely used reference for machine-learning-based analyses of neuroimaging data (the introductory paper by Abraham et al., 2014, has been cited ~1200 times). Scikit-learn is co-developed by the INRIA Parietal team which is also hosted at Neurospin, and we have used it ourselves in multiple previous publications (e.g., Eger et al., 2015 *Cerebral Cortex*, Lasne et al., 2018, Castaldi et al., 2019). Scikit-learn does not provide dedicated functionality for RSA, which is why, rather than implementing our own version of it, for these parts we relied on CosmoMVPA (<http://www.cosmomvpa.org/>), which does in particular provide an option for the multiple regression-based analyses on distance matrices that we present here. We found this toolbox well-structured and documented and easy to use (and have also used it for the RSA analyses in Castaldi et al., (2019)). It is not very clear how the different analyses performed in our manuscript could be implemented using the decoding toolbox (which, as it seems to us at least, is rather lacking clear documentation and tutorials in comparison with the two options that we used). We believe having reported all the necessary details on how our analyses

were conducted, and think that the software used for these should remain our choice, especially since we are using options that are publicly available and well established in the field.

Significance testing for the cross-decoding analysis could be added.

Maybe you could specify your FDR correction method in a bit more detail, e.g. “Multiple comparison correction was based on FDR across subregions within lobe”, by motivating why it is done within lobe and providing numbers of the amount of regions and cut-off significance values as well as p-values where possible.

We have redone the statistical analysis for cross-decoding, and its correlation with behavior, with correction for multiple comparisons across all ROIs (32 regions in total, 16 within each lobe). The resulting statistics tables are now reported as in the Supplementary Materials (Supplementary Tables 3 and 4).

Related to this, I am not sure if it is helpful to report in line 179 uncorrected $p < 0.05$ significance – maybe see also comment on FDR correction, it would help to see where FDR correction is doing the cut-off to evaluate this.

We have removed this uncorrected significance from the results.

What does “multiple regression” (line 652) refer to? Please describe this in more detail

This is referring to the RSA analysis with multiple regression of the 4 predictor matrices (sample, operation, operand, result) on fMRI distance matrices, as now explicitly spelled out.

With regards to Tables and Figures:

Organization of Table1: Typically the anatomical regions are written on the left.

We have reorganized this and other new tables so that the anatomical regions appear on the left.

The authors use different types of surface renderings in Figure 1 and for the presentation of the ROIs. Would it be possible to use the same display to make the findings more accessible? It might be much to ask for – and it might be a bit tricky to do such a visualization, but maybe one could also overlay the borders of the surface parcellation (from Figure 3) with the results of the searchlight analysis in Figure 2. Alternatively, one could try to plot the results presented in Figure 3 into the same kind of surface rendering as presented in Figure 2 (by dropping the error bars, and coloring each ROI according to the beta weight value).

We are now overlaying the borders of the HCP-MMP1 surface parcellation onto the flattened surfaces displaying the searchlight analysis results. For display of the regions included in the ROI analysis, we introduced a flat map as well, with magnification of the critical parts containing the ROIs. We found coloring the ROIs by beta weight in-satisfactory since it cannot provide information on the distribution which was requested by the journal (therefore now the violin plots).

Figure 1: the table might be very small in the final print. Was the ITI fixed to 7.2 sec? Why? The correspondence of the RSA lines and columns could be made more clear

We have enlarged the mentioned table on the right and its font size.

Total duration of each trial was 20 s, variable durations would in fact complicate and could produce counterintuitive results in the analysis by FIR model if in some cases for the chosen window length FIR bins of different trials overlap and in others not. Of the total 20 s, on standard trials 12 s corresponded to sample / operation presentation and delay period and 8 s to the probe and response period (resulting in 7.2 s of blank screen after stimulus and response prompt). On a minority of catch trials, the delay period was unpredictably shortened (see page 11). In these cases, the blank period after the probe was lengthened so that the total trial was still 20 s. We have added a note in the figure legend stating that this illustration refers to the standard trial type.

The predictor matrices and beta weight illustration of the RSA in part B of the figure has been rearranged to make the correspondence more intuitive.

Figure 2: The authors could additionally report contrasts of e.g. early > late or early > (mid & late) etc. (maybe in the supplement)

In response to this suggestion, we are now reporting differential contrasts between different parts of the delay period for the searchlight pattern analysis: mid > early, late > early, late > mid and mid & late > early comparisons. Results are reported in Supplementary Figure 2 and Supplementary table 2.

Figure 3: All x-axis for the beta weights should be scaled in the same way, as otherwise one is tempted to compare the effect sizes between the sample and results, which are not scaled in the same way.

The axes are now scaled equally for the effects of sample and result numerosity, and the different ROI groups.

Figure 4A: I do not understand from the figure nor from the text how the visualization of the ROIs in Figure 4A relates to the plots below. Three of the scale bars can be removed and the four matrices can be aligned so that the labelling of the x- and y-dimension is only provided once. These types of plots are typically plotted in a way so that the diagonal is from the bottom left to the top right (See EEG-decoding literature), and I would consider it helpful to stick to this way of visualization. It remains unclear for which ROI this is plotted?

Panel A shows all ROIs for the results in C. B is showing the decoding results (within sample, result, and for cross-decoding between them) for one example ROI (MIP), which is now stated in the figure legend. The direction of visualization has been changed in the requested way.

Figure 4B: Plots could be made more efficient by dropping redundant axis labelling.

The redundant labels in the correlation plots have been dropped.

There are 5 figures but only 4 figure legends. Figure 5 (I assume the supplementary one): As for Figure 3, all axis should be plotted on the same scaling.

The previous Figure 5 is now Supplementary Figure 5 and has its axes scaled equally as well.

Reviewer #3 (Remarks to the Author):

Summary

In this study Czajko, Vignaud and Eger investigate the neural representation of internally generated outcomes of approximate (i.e. non-symbolic) multiplication and division problems by means of high-field neuroimaging. After having been presented with a set of dots (sample) participants were prompted with a multiplier/divisor in symbolic format (either 2 or 4, operand). After the delay period during which they were asked to maintain the result in working memory, participants were probed with a comparison numerosity (probe) that they had to compare to the maintained outcome. Using a searchlight algorithm for detecting BOLD signatures of operands, operation and outcome in a 'time-resolved' (i.e. TR-wise) fashion, the authors detected marked effects of the sample numerosity in bilateral parietal areas along the intraparietal sulcus and "lateral prefrontal and occipito-temporal brain regions". In time bins immediately preceding the probe, signatures of the outcome were observed in left middle frontal gyrus, the right inferior precentral sulcus, the left angular gyrus and the right medial parietal lobe. Additional analyses revealed signatures of the result in superior wall of the IPS (regions IP1 and MIP) in the same time window. Common coding of (early) sample and (late) result numerosity was observed in MIP. The decoding accuracy in this analysis correlated with the behaviorally observed JND across participants.

These results are interpreted as evidence for a transformation of sample into result quantities in dorsal stream sensory-motor integration regions. The generated results would further be maintained for task purposes in higher-level regions in a format possibly detached from sensory evoked representations, involving angular gyrus, middle frontal gyrus and precentral sulcus.

Evaluation

This study tackles an important question in numerical cognition. What cortical circuits are involved in generating and maintaining the outcome of arithmetic problems and to what extent do these circuits overlap with those involved in decoding the perceptual input numerosity? This question – albeit being central to much of mathematical cognition – has remained unanswered so far. Most work in this domain has been devoted to the deciphering of the neural codes underlying perceived quantities. Hence, the results present an important step towards a comprehensive understanding of the processing steps that are involved in solving arithmetic problems. The study is expertly designed and analyzed. The manuscript is overall well written but some formulations may merit the authors' attention. Overall, I do not see any reasons that would prevent this study from ultimately being published. I think it would be well received in the community. The authors may find the following comments helpful in further improving the manuscript.

We thank the reviewer for the positive evaluation and the constructive comments.

Comments

The authors chose an intermediate step by investigating non-symbolic (approximate) arithmetic - a choice presumably driven by the difficulty of decoding symbolic quantities. It may nevertheless be helpful to inform the reader why the current design – in particular the non-symbolic format – was chosen over more common symbolic arithmetic problems. In this context, a distinction that is often found in the literature may be discussed: approximate versus exact calculation.

What implications do the current results have for symbolic calculations?

Indeed, our choice of using a non-symbolic calculation task is related to the fact that previous studies showed that brain representations of individual symbolic numbers are more difficult to distinguish than dot numerosities on the basis of their evoked fMRI activity. In addition, we think that the need for actually manipulating some form of quantity is higher in non-symbolic calculation compared to some symbolic calculation paradigms where overlearned outcomes might simply be retrieved from verbal memory. Given the use of non-symbolic numbers, our paradigm indeed corresponds to an

approximate calculation task, which we are now explicitly referring to in the revised manuscript (e.g., title and abstract) and we explain the reason for choosing a non-symbolic task at the end of the introduction (page 4).

Since we did not test calculation on symbolic numbers, we can only speculate on the implication of our results for this case. It is possible that in cases where a symbolic task would encourage the use of an approximate strategy as here with non-symbolic quantities, result representations could be similar as observed in our case, whereas in cases where the symbolic task would rely on retrieval of precise, language-based representations from memory, results would be different. These are important questions for future studies as we now point out at the end of our discussion (page 12).

The non-symbolic format may afford visual strategies to solve the given problem. That is, participants may imagine the individual dots split in two or four for multiplication or to merge for division. One may hence argue that no numerical representation is required to solve the task because participants may rely on non-numerical features. While I am not arguing that this is necessarily the strategy that participants adopted because numerosity is a salient stimulus dimension that is readily used by participants, it may still be interesting to discuss this possibility.

We cannot exclude that participants did rely on some form of visual imagery to solve the task (and we think that this is true for most if not all calculation tasks on non-symbolic stimuli). However, the use of different stimulus sets (where dots were shown in either a large or a small field area unpredictably varying between sample and probe, in addition to the fact that item size was also variable within set) does in our opinion make very simple non-numerical visual strategies such as reliance on either splitting the area or dividing the density insufficient. We therefore consider it likely that if subjects did not rely on entirely non-visual (abstract) representations of number, they still relied on visual “numerical” representations (that is, on representations of discrete numbers of items, as opposed to their non-numerical characteristics). We are now briefly discussing this in the manuscript (page 9).

Did authors adopt a more classical analysis approach (i.e. convolving the signal with a canonical HRF) and tried to identify regions that show numerosity-specificity or even place coding as shown for single parietal and prefrontal neurons in NHPs?

We had not originally performed such more classical analyses since in our design using a calculation task with several stimulus and task related components over time a canonical HRF shape did not appear the most obvious choice, and we expected result-related activation to be subtle and not necessarily easily detectable at the level of individual voxels. In response to the reviewer’s request, we have now fitted an additional first-level model using a canonical HRF and the following predictors: four predictors for the four different sample numerosities, two predictors for the two operands (2 and 4), two predictors for the two operations (multiplication and division), four predictors for the four possible results, and eight predictors for possible probe numerosities (split into smaller and larger for each possible result). We then conducted second-level analyses for the contrasts corresponding to a parametric increase, and to enhanced responses for 6-other, 12-other, 24-other, and 48-other, for both sample and result numerosities. Results are reported in Supplementary Figure 1 and Supplementary Table 1, and mainly show enhanced activations in intra-parietal regions for sample 6, and in early visual cortex for sample 48 (in addition to some effects in the right angular gyrus for result 6 and 24). We would like to point out though, that since these analyses rely on an assumed shape of the hemodynamic response, if the truly evoked activations for the different components of the task should be different from this shape, there is a risk that parts of the variance

could be assigned to the wrong predictor. Our main analyses using FIR models which are assumption free in terms of the shape of the hemodynamic response, in combination with including all predictors to explain activation pattern dissimilarity at each possible time point, where each predictor should yield betas corresponding to the variance it can explain on top of all the others, should be better suited to this kind of situation. We therefore report the requested HRF based analyses for completeness, but do not place strong conclusions on their results.

Moreover, given that we are dealing with fMRI data and that the design included only four different numerosity levels, we think we are not well placed to draw conclusions about tuning functions such as place or summation codes, which even in single units sometimes may be non-trivial to distinguish (see e.g., Chen and Verguts, 2013, *Front Hum Neurosci*). Previous fMRI studies that found numerosity tuned or monotonic responses of individual voxels in fMRI (e.g., Harvey et al., 2013, *Science*, Paul et al. 2022, *Nat Commun*), used population receptive field mapping methods in individual subjects, an approach which we do not currently have established in our lab. These studies also maximized detection power by dense stimulus presentations (contrasting with the rather sparse and comparably few stimuli here), and their designs were comparably simple (not requiring to disentangle other task related components from the mere evoked numerosity activations as here). Nevertheless, numerosity tuned responses even under these conditions were detected much less frequently for numerosities larger than 7 compared to smaller numerosities (Cai et al., 2021, *Nat Commun*). Therefore, successful application of the same approach to our specific situation would be anything else but trivial, and if at all possible, we think it cannot be within the scope of the present contribution.

Plotting the time-course in ROIs with significant decoding would be informative, too.

We are now providing plots of the average activation time courses for all the ROIs used in the RSA and decoding analyses (Supplementary Figures 3 and 4).

My copy of the manuscript contained a figure 5 for which I found no caption and that was not referenced in the text body.

This figure is now integrated into the Supplementary Materials (Supplementary Figure 5).

Authors report overall accuracies of ~75%. It would be informative to report the accuracies as a function of ratio that allowed them to estimate JNDs. Also, did authors analyse the effect of ratio on BOLD?

Accuracies as a function of ratio are now reported on page 5.

We did not originally think that the BOLD effects due to probe ratio would be of much interest, since they should only occur after probe presentation while we are focusing in our result related analyses on the preceding delay period, where the probe to follow cannot yet be predicted. We have now fitted an additional FIR model where the conditions are redefined as a function of ratio, since this could serve as a check for whether effects related to it are indeed absent from the data during other periods of the trial (i.e., do not affect the following trial's delay period in any systematic way). Results confirmed the expectation that significant effects of probe ratio (close > far ratio), corresponding to a classical comparison distance effect as often reported in the literature, showed the expected results

only during the late probe interval (4-8 s after probe presentation). For now, we are showing these results below, but can also add them to the Supplementary Materials if judged sufficiently relevant.

Figure 1 : Surface based group analysis (N=17) of differential effects of close > far probe ratio during the late probe interval (one-sample t-test, corrected for multiple comparisons by permutation at cluster level, p_{FWE} < .05, cluster forming threshold $p < .01$).

Table 1: Cluster summary table for the surface-based group analysis (N=17) conducted on the univariate contrast of probe ratio (close > far).

For each cluster surviving $p_{FWE} < .05$ (corrected at cluster level by permutation methods with cluster forming threshold $p < .01$) the table reports: the cluster label for the activation maximum (as defined by the anatomical labels from the Destrieux Atlas), the maximum $-\log_{10}(p)$ value in the cluster (Max), the cluster surface area in mm² (Size), the MNI coordinates of the maximally activated vertex within each cluster (MNI X, Y, Z), the cluster-wise p-value of each cluster (CWP), and the number of vertices included in each cluster (NVtxs). No above threshold clusters were detected for the early, middle and late delay period, nor for the early probe period.

Ratio (close > far) – late probe period							
Left hemisphere							
Cluster Label	Max	Size (mm ²)	MNI X	MNI Y	MNI Z	CWP	NVtxs
S_intrapariet_and_P_trans	5.46	513.0	-33	-46.6	44	0.0223	1408
G_and_S_cingul-Mid-Ant	4.89	465.4	-10.8	15.7	43	0.0302	900
S_intrapariet_and_P_trans	3.76	416.6	-21.5	-72.3	34.8	0.0357	857
G_precentral	5.95	406.5	-53.6	4.7	25.9	0.0376	869
S_front_sup	4.18	404.3	-21.1	-4.6	48.3	0.0384	890
Right hemisphere							
Cluster Label	Max	Size (mm ²)	MNI X	MNI Y	MNI Z	CWP	NVtxs
G_and_S_cingul-Mid-Ant	5.2302	756.59	13.9	25.4	27.8	0.0148	1588
S_front_inf	5.2304	510.1	37.5	21	22.7	0.0282	1149
S_intrapariet_and_P_trans	3.8723	493.39	18.4	-66.2	47.6	0.0306	858
S_precentral-sup-part	4.5081	448.14	24.4	-6.1	45.5	0.0357	1032
S_intrapariet_and_P_trans	3.7602	417.24	31.9	-41.2	38.5	0.0400	1106

When inspecting figure 4, I was wondering how authors would interpret the cross-decoding between early sample and late result that appears as 'red blob' next to (sample-result) and below (result-sample) the highlighted time window that authors focus on?

A second question that jumps to my mind is why this later time window is the same where authors observe the highest accuracy in decoding the result numerosity (result-result).

This mentioned red blob corresponds to cross-decoding observed between the sample numerosity and the internally generated result in combination with the stimulus evoked activity of the probe numerosity (which is averaged here across the different probe ratios used which for each possible result differed by the same set of ratios towards the smaller or larger across trials). The likely reason why decoding of the result appears most pronounced in this later time period is that decoding during this time period is mostly driven by the stimulus-evoked activity of the probe. In sum, while it is not a result with interesting theoretical implications per se, this generalization between sample and result in the latest time window provides a good sanity check for the analysis procedures, demonstrating that they can in principle identify correspondence in contents of representations across different moments of the trial time course.

We have added an explanation regarding the correspondence of the mentioned cross-decoding performance in the late time period to the figure legend.

Minor comments

Line 64: "humans has proceeded to characterizing the representations / neural codes of numerical information" Please provide a verbal conjunction instead of "/"

170/171: "Results showed that information *about* the sample number was widely *distributed* in parietal cortex (Figure 3 A),

181: Overall, these ROI analysis result*s*

251: This region* is also the one

Thanks for spotting these diverse errors, they have been corrected.

REVIEWERS' COMMENTS

Reviewer #1 (Remarks to the Author):

I am now happy with the revised manuscript. I have no further suggestions for improvements. Very nice work!

Reviewer #2 (Remarks to the Author):

The authors answered all my points satisfactorily.

Reviewer #3 (Remarks to the Author):

I thank the authors for their responses to my comments that have all been addressed and clarified. I think the manuscript reads very well and represents an important step forward in elucidating how the brain accommodates mental arithmetic. The discussion of the results is balanced and well-thought.

My only comment refers to the ratio effect on the probe period which I think is an important sanity check and should appear in the supplementary material but I will leave this decision to the authors. The manuscript can be published with or without this piece of information (which would also be potentially interesting for future meta analyses).

minor:

p. 8: "The use*d* of long delays [...]". It should be "use".

Response to reviewers - Czajko et al. - revision 2

Reviewer #3 (Remarks to the Author):

I thank the authors for their responses to my comments that have all been addressed and clarified. I think the manuscript reads very well and represents an important step forward in elucidating how the brain accommodates mental arithmetic. The discussion of the results is balanced and well-thought.

My only comment refers to the ratio effect on the probe period which I think is an important sanity check and should appear in the supplementary material but I will leave this decision to the authors. The manuscript can be published with or without this piece of information (which would also be potentially interesting for future meta analyses).

The results concerning the ratio effect in the probe period have been integrated into the Supplementary Materials.

minor:

p. 8: "The use*d* of long delays [...]". It should be "use".

Thanks for spotting this error which is now corrected.